# PROREGEN: PROGRESSIVE RESIDUAL GENERATION UNDER ATTRIBUTE CORRELATIONS

**Ruby Shrestha, Ajay Gopi, Casey Meisenzahl, Bipin Lekhak & Linwei Wang**
Rochester Institute of Technology, Rochester, NY, USA
`rs9466@rit.edu`

## ABSTRACT

Attribute correlations in the training data will compromise the ability of a deep generative model (DGM) to synthesize images with under-represented attribute combinations (*i.e.,* minority samples). Existing approaches mitigate this by data re-sampling to remove attribute correlations seen by the DGM, using a classifier to provide *pseudo-supervision* on generated counterfactual samples, or incorporating inductive bias to explicitly decompose the generation into independent sub-mechanisms. We present ProReGen, a *progressive residual generation* approach inspired by the classical Robinson's transformation, to partial out from an image attribute $\mathbf{x}_2$ its component $m(\mathbf{x}_1)$ that is predictable by other image attributes $\mathbf{x}_1$, and the residual $\gamma = \mathbf{x}_2 - m(\mathbf{x}_1)$ that is not. This simplifies the problem of learning a DGM $g(\mathbf{x}_1, \mathbf{x}_2)$ conditioned on correlated inputs, to learning $\tilde{g}(\mathbf{x}_1, \gamma)$ conditioned on orthogonal inputs. It further allows us to progressively learn $\tilde{g}$ by first shifting the burden to abundant majority samples to learn $\tilde{g}(\mathbf{x}_1, \gamma = 0)$, and then expanding it with additional layers $g_{\text{res}}$ to resolve its difference to $\tilde{g}(\mathbf{x}_1, \gamma)$ using residual attribute $\gamma$ on limited minority samples. On three benchmark datasets with curated varying strengths of attribute correlation and one dataset with natural attribute correlation, we demonstrate that ProReGen—with input orthogonalization and progressive residual learning—improved the correctness of minority generations compared to existing strategies.

## 1 INTRODUCTION

Attribute correlations are not uncommon in observed image datasets. Some may be a natural manifestation of underlying causal relations, *e.g.,* the object in an image determining the background (Sagawa et al., 2019). Some may reflect bias in data curation, *e.g.,* collecting patient data from those who already received treatment (Wang et al., 2017). Regardless of the mechanisms, attribute correlations can induce unintended consequences in deep neural networks (DNN) training.

In the context of discriminative (*e.g.*, classification) DNNs, this phenomenon is widely discussed, often under the concept of *spurious correlations* or *short-cut learning* (Ye et al., 2024). In the context of deep generative models (DGM), such discussion is comparatively less structured and scatters across a variety of topics. On one hand, DGMs are used in many domains to synthesize under-represented image examples—those with image attributes that do not comply with the observed correlation, *e.g.,* for explaining whether a DNN classifier has captured correlated features for decision making (Rodríguez et al., 2021), or for augmenting training data to mitigate correlations (Kim et al., 2021). On the other hand, several evaluation studies (Träuble et al., 2021; Bose et al., 2022) have shown that naively-trained DGMs would capture latent attribute correlations from training data (Träuble et al., 2021; Bose et al., 2022). How does this impact the synthesis of under-represented samples, and to what extent could it be mitigated? Answers to these questions remain open.

Consider two sets of image attributes $\mathbf{x}_1$ and $\mathbf{x}_2$ (both can be multi-dimensional) that exhibit a correlation in observed data. Consider the goal of learning a DGM $g$ conditioned on these attributes to generate an image $\mathbf{y}$ as $\mathbf{y} = g(\mathbf{x}_1, \mathbf{x}_2)$. For the function $g$ to generate with different combinations of $\mathbf{x}_1$ and $\mathbf{x}_2$ values, it is important for $g$ to correctly model the mechanisms, $g_1$ and $g_2$, through which $\mathbf{x}_1$ and $\mathbf{x}_2$ influences $\mathbf{y}$ separately. Unfortunately, due to the observed $\mathbf{x}_1$-$\mathbf{x}_2$ correlation, $g_1$

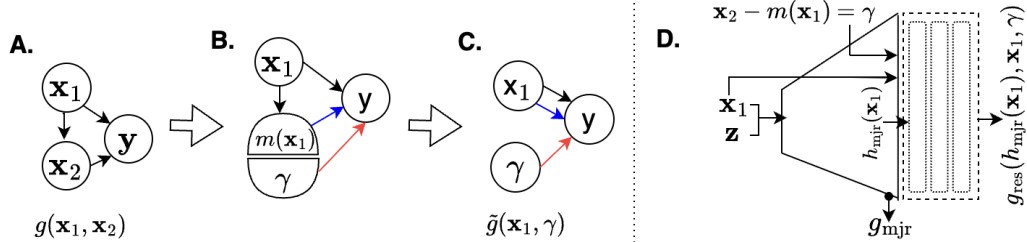

Figure 1: A-C: Overview of the Robinson's partialling-out approach in motivating learning $g(\mathbf{x}_1, \mathbf{x}_2)$ as $\tilde{g}(\mathbf{x}_1, \gamma)$. D: Conceptual illustration of ProReGen to realize function $\tilde{g}(\mathbf{x}_1, \gamma)$ by progressively learning $g_{\mathrm{mjr}}(\mathbf{x}_1) \coloneqq \tilde{g}(\mathbf{x}_1, \gamma = 0)$ from majority samples, followed by learning $g_{\mathrm{res}}(\mathbf{h}_{\mathrm{mjr}}(\mathbf{x}_1), \mathbf{x}_1, \gamma)$ on minority samples to resolve the difference to $\tilde{g}(\mathbf{x}_1, \gamma)$ with $\gamma$.

and $g_2$ can only be separately observed in the small number of samples where such correlation does not hold. We stress this as a fundamental challenge for learning a DGM under attribute correlations.

Existing strategies to address this challenge are limited. Re-sampling is a simple approach to balance training samples (Monteiro et al., 2022), essentially up-weighting under-represented samples where $g_1$ and $g_2$ can be separately observed. Alternatively, inductive bias has been introduced to explicitly decompose $g$ into independent mechanisms $g_1$ and $g_2$, which requires prior knowledge about how $\mathbf{x}_1$ and $\mathbf{x}_2$ may influence $\mathbf{y}$ differently (*e.g.,* object shape *vs.* texture *vs.* background) (Sauer & Geiger, 2020). Finally, to go beyond the limits of *factual* under-represented samples, *pseudo-supervision* on generated counterfactual images has been presented, typically realized by using a classifier to recognize feature attributes in the generated images (Kocaoglu et al., 2017; Ribeiro et al., 2023; He et al., 2019). However, since the classifier is trained under the same attribute correlations, its ability to correctly recognize these attributes is likely compromised – how does this impact its validity to *supervise* the generation of under-represented counterfactual images has not been well understood.

In this paper, we take a fundamentally different perspective to address the challenge of modeling $g(\mathbf{x}_1, \mathbf{x}_2)$ under attribute correlations, inspired by the classical Robinson's *partialling-out* transformation (Robinson, 1988). While details of this concept will be introduced in Section 3, Fig. 1 illustrates its core concept in the context of modeling $g(\mathbf{x}_1, \mathbf{x}_2)$. Consider the causal graph in Fig. 1A with a causal direction assumed between $\mathbf{x}_1$ and $\mathbf{x}_2$. Instead of attempting to model the independent causal mechanisms $g_1$ and $g_2$ in the presence of such correlations, we decompose $\mathbf{x}_2$ into $E[\mathbf{x}_2|\mathbf{x}_1]$ that can be predicted by $\mathbf{x}_1$, along with a residual $\gamma$ that cannot, *i.e.,* $\mathbf{x}_2 = E[\mathbf{x}_2|\mathbf{x}_1] + \gamma$ (Fig. 1B). With this, instead of modeling $\mathbf{y} = g(\mathbf{x}_1, \mathbf{x}_2)$ as a composition of $g_1$ and $g_2$ as in Fig. 1A, we model it as $\mathbf{y} = \tilde{g}(\mathbf{x}_1, \gamma)$ as in Fig. 1C: the effect of $\mathbf{x}_1$ on $\mathbf{y}$ now absorbs the effect from $E[\mathbf{x}_2|\mathbf{x}_1]$, the component of $\mathbf{x}_2$ that can be predicted from $\mathbf{x}_1$; – we referred to this as *correlated effect*; as such, the effect of $\gamma$ on $\mathbf{y}$—the part of $\mathbf{x}_2$'s influence on $\mathbf{y}$ *that cannot be explained by* $\mathbf{x}_1$—is *partialled out*: we refer to this as *residual effect*. In this new causal graph (Fig. 1C), instead of attempting to recover two independent mechanisms from correlated inputs $\mathbf{x}_1$ and $\mathbf{x}_2$, we transform the problem into learning independent *correlated* and *residual effects* from two independent inputs $\mathbf{x}_1$ and $\gamma$.

To design a DGM based on Fig. 1C, we note that $\tilde{g}(\mathbf{x}_1, \gamma)$ at $\gamma = 0$, *i.e.,* $E[\mathbf{y}|\mathbf{x}_1, \gamma = 0]$, can be estimated from abundant samples whose attribute values meet the observed correlations (henceforth referred to as *majority samples*). $\tilde{g}(\mathbf{x}_1, \gamma)$ at $\gamma \neq 0$, on the other hand, can only be estimated from a limited number of samples where such correlation does not hold (henceforth referred to as *minority samples*). To shift the primary burden of learning $\tilde{g}$ to majority samples, therefore, we further decompose the learning of $\tilde{g}(\mathbf{x}_1, \gamma)$ into the learning of $\tilde{g}(\mathbf{x}_1, 0)$ at $\gamma = 0$ using majority samples, and use minority samples to make up the difference between $\tilde{g}(\mathbf{x}_1, 0)$ to $\tilde{g}(\mathbf{x}_1, \gamma)$. This results in our progressive residual effect generator (ProReGen) that is progressively learned in two stages as outlined in Fig. 1D. ProReGen as described is expected to have two major benefits. First, the orthogonalization of the inputs $\mathbf{x}_1$ and $\gamma$ helps separate the learning of their independent effects on $\mathbf{y}$. Second, with the progressive expansion from $g_{\mathrm{mjr}}$ to $g_{\mathrm{res}}$, the challenge of learning to generate under attribute correlations is reduced to learning the residual between $\tilde{g}(\mathbf{x}_1, \gamma = 0)$ and $\tilde{g}(\mathbf{x}_1, \gamma)$. We instantiate the concept of ProReGan in conditional-VAEs, -GANs, and -diffusion models (DMs). On three benchmark datasets with varying strengths of attribute correlations and one dataset with natural attribute correlations, we experimentally demonstrate the improved performance of ProRe-

Gen in generating *correct* minority samples compared to naive DGMs or those strengthened with re-weighting of factual samples or pseudo-supervision of generated samples.

## 2   RELATED WORKS

Several domains find uses for synthesizing minority images. For instance, to explain if a classifier has captured attribute correlations from its training data, one can test if it is able to guide a pre-trained DGM to generate minority images (Rodríguez et al., 2021; Jeanneret et al., 2022). In addition to explaining, synthesized minority images can also be used for augmenting and removing attribute correlations in training data (Goel et al., 2020; Kim et al., 2021). While naively-trained DGMs have been used for such syntheses (Rodríguez et al., 2021; Jeanneret et al., 2022), there is an increasing attention on the impact of attribute correlations on DGM training and potential mitigation solutions.

**Re-weighting of factual samples:** The concept of *simulated intervention* was presented in Monteiro et al. (2022) to re-sample data according to the marginal distribution of image attributes, to effectively remove attribute correlations seen by the DGM. This is essentially similar to re-weighting, where minority samples are up-weighted in their contribution to the training signals. This approach is ultimately affected by the quantity and diversity of factual minority samples.

**Pseudo-labeling of counterfactual generations:** Instead of relying on factual minority samples, an alternative is to provide some *pseudo-supervision* to encourage the DGM to generate *counterfactual* images with intended feature attributes. This is often achieved by leveraging another DNN classifier, often trained from the same data as the DGM, to provide supervisory signals by recognizing the attributes of generated images (Kocaoglu et al., 2017; Ribeiro et al., 2023; He et al., 2019). However, because the DNN classifier is also subject to attribute correlations in the training data, their reliability in correctly recognizing these attributes is questionable. How does this affect the supervisory signal it provides to the DGM's counterfactual generations has not been systematically investigated.

**Inductive bias to decompose generation mechanisms:** An entirely different approach is to incorporate inductive bias about the mechanisms under which different attributes contribute to generated images (Sauer & Geiger, 2020; Park et al., 2020) In (Sauer & Geiger, 2020), for instance, the image generation process is decomposed into independent shape, texture, and background mechanisms. While this approach tends to be highly successful when assumptions of the underlying generation mechanisms are met, it does require prior knowledge for the design of such inductive bias.

**Disentanglement under attribute correlations:** More broadly, several evaluation studies have shown that naively-trained DGMs would capture attribute correlations in entangled representations (Träuble et al., 2021) and even inherit the associated causal directions (Bose et al., 2022). To learn disentangled represenations from correlated attributes, existing works in VAEs *e.g.*, encourage pairwise factorized support by minimizing a hausdorf distance (Roth et al., 2023), minimize conditional mutual information between subspaces with respect to categorical variables (Funke et al., 2021), or extend property-controllable VAEs to disentangle groups of properties while allowing correlations within the group. Similarly, in GANs, works exist to navigate GAN's latent space to disentangle otherwise entangled attributes, *e.g.*, by latent optimization (Li et al., 2020) or leveraging gradien information (Chen et al., 2022). In general, these works focus on disentangled representations or image manipulation, rather than generation of minority samples as ProReGen.

**Generative modeling under attribute imbalance:** Finally, ProReGen is related to generative modeling under attribute imbalance (Georgopoulos et al., 2020; Zhao et al., 2018; Haider et al., 2025) although the challenges they tackle differ: while imbalance presents a general generalization challenge for the model to represent rare attributes, correlated attributes require methods to explicitly handle dependence among co-occurring attributes as a specific form of distribution shift.

ProReGen represents a completely different approach to these existing works. Inspired by the classical Robinson's partialling out approach, ProReGen tackles the challenge of attribute correlations at its core by first recasting the DGM from being conditioned on correlated attributes to orthogonal attributes. By a progressive expansion design, it further leverages majority training samples to reduce the problem of learning minority generation to learning its residual to majority generation. The latter concept is related image translation models that transform a factual majority sample to a minority counterfactual (Kim et al., 2021; Goel et al., 2020; An et al., 2022). Because these models are intended only for translating factual samples, they are out of the scope of our consideration.

## 3   PRELIMINARY: ROBINSON'S PARTIALLING-OUT TRANSFORMATION

Consider a partial linear equation $E[y|x_1, x_2] = \theta(x_1) + \beta x_2$, where $x_2 = m(x_1) + u$, Robinson's transformation in (Robinson, 1988) decomposes the original equation into:

$$E[y|x_1, x_2] = \theta(x_1) + \beta * (m(x_1) + u) = E[y|x_1] + \beta * (x_2 - m(x_1)) \tag{1}$$

where $E[y|x_1] = \theta(x_1) + \beta * m(x_1)$. Effectively, instead of describing the separate effect of $x_1$ and $x_2$ on $y$ as dictated the original equation, their effect is decomposed into two orthogonal components: 1) $E[y|x_1]$ that describes the combined effect of $x_1$ and $m(x_1)$ on $y$—the latter absorbing the effect of $x_2$ on $y$ that can be predicted by $x_1$; and 2) the *residual* effect of $\beta * (x_2 - m(x_1))$ from $x_2 - m(x_1)$—the effect on $y$ from the residual of $x_2$ that cannot be predicted from $x_1$.

In the original paper (Robinson, 1988), the goal of this decomposition is to estimate parameter $\beta$, which arrives at the classical *residual-on-residual* least-square fitting of $\beta$ via: $E[y|x_1, x_2] - E[y|x_1] = \beta * (x_2 - E[x_2|x_1])$. The resulting estimator for $\beta$ can further be shown to meet the *Neyman's orthogonality condition* such that it is insensitive to perturbations in the estimator for $E[y|x_1]$ or $E[x_2|x_1]$ (Robinson, 1988; Chernozhukov et al., 2018). This decomposition has also served as the basis for the R-learner (R for Robinson) that extended the approach to estimating the function $\beta(x_1)$ instead of the low-dimensional parameter $\beta$ (Nie & Wager, 2021).

In contrast, we use this transformation as the foundation inspiring the design of an image-generating DGM, where $y$ is high-dimensional and its relation with $x_1$ and $x_2$ is highly nonlinear. Note that, in the original formulation, the causal direction $x_1 \rightarrow x_2$ depends on the target parameter $\beta$; in our setting where $y$ as a function of $g$ remains the primary interest, the assumed causal direction will influence the decomposition, but not the general modeling strategy. The difficulty of the residual generation task however may change with this direction, which we will empirically examine in Section 5.5. Below we continue our discussion with $x_1 \rightarrow x_2$ without loss of generality.

## 4   METHODOLOGY

### 4.1   PROREGEN: PROGRESSIVE RESIDUAL GENERATION

Consider a conditional-DGM $\mathbf{y} = g(\mathbf{z}, \mathbf{x}_1, \mathbf{x}_2)$ underlying observed image data, where a correlation exists between image attributes $\mathbf{x}_1$ and $\mathbf{x}_2$, and $\mathbf{z}$ represents latent variables not included in $\mathbf{x}_1$ and $\mathbf{x}_2$. Built on Robinson's partialling-out approach, as illustrated in Fig. 1A-C, we first cast the problem of learning $\mathbf{y} = g(\mathbf{z}, \mathbf{x}_1, \mathbf{x}_2)$ with correlated inputs, to learning $\mathbf{y} = \tilde{g}(\mathbf{z}, \mathbf{x}_1, \gamma)$ with independent inputs where $\gamma = \mathbf{x}_2 - m(\mathbf{x}_1)$ represents the residual in $\mathbf{x}_2$ that cannot be predicted by $\mathbf{x}_1$. Because $\tilde{g}(\mathbf{z}, \mathbf{x}_1, \gamma = 0)$ is described by majority samples *vs.* $\tilde{g}(\mathbf{z}, \mathbf{x}_1, \gamma)$ at $\gamma \neq 0$ by a small number of minority samples, we further design a progressive learning strategy to shift the burden of learning $\tilde{g}$ mostly to the learning of $\tilde{g}(\mathbf{z}, \mathbf{x}_1, \gamma = 0)$ using majority samples, and using minority samples to only resolve its difference to $\tilde{g}(\mathbf{z}, \mathbf{x}_1, \gamma)$ with $\gamma$. This gives the foundation of ProReGen:

$$\mathbf{y} = \tilde{g}(\mathbf{z}, \mathbf{x}_1, \gamma) \approx g_{\text{res}}(\mathbf{h}_{\text{mjr}}(\mathbf{x}_1), \mathbf{x}_1, \gamma), \quad \text{where} \quad \gamma = \mathbf{x}_2 - m(\mathbf{x}_1) \tag{2}$$

where $\mathbf{h}_{\text{mjr}}$ is the feature map of $g_{\text{mjr}} := \tilde{g}(\mathbf{z}, \mathbf{x}_1, \gamma = 0)$ before the final activation layer. Equation (2) includes three main components progressively learned in two stages, as illustrated in Fig. 1D:

- In stage-I, from a large number of majority samples, we learn an attribute predict function $\mathbf{x}_2 = m(\mathbf{x}_1)$ to approximate $E[\mathbf{x}_2|\mathbf{x}_1]$, and a generative model $g_{\text{mjr}}(\mathbf{x}_1)$ to approximate $\tilde{g}(\mathbf{x}_1, \gamma = 0)$, the latter effectively describing the generation when $\mathbf{x}_2 = m(\mathbf{x}_1)$, *i.e.,* for a *majority* sample.

- In stage-II, using available minority samples, we expand the generator with additional layers, $g_{\text{res}}$, to resolve the residual between $\tilde{g}(\mathbf{x}_1, \gamma)$ and $g_{\text{mjr}}(\mathbf{x}_1)$ with the residual $\gamma$ partialled out from $\mathbf{x}_2$. Effectively, we approximate $\tilde{g}(\mathbf{x}_1, \gamma)$ by $g_{\text{res}}(\mathbf{h}_{\text{mjr}}(\mathbf{x}_1), \mathbf{x}_1, \gamma)$, with $\mathbf{h}_{\text{mjr}}$ defined above.

This takes form of a progressive-DGM, where a backbone $g_{\text{mjr}}$ is first learned on majority samples and then expanded on minority samples. This concept is agnostic to the type of DGMs: below, we describe its instantiations on conditional-VAEs (c-VAEs), -GANs (c-GANs), and -DMs (c-DMs).

### 4.2   PROREGEN-VAE

**Stage I:** To learn $g_{\text{mjr}}(\mathbf{z}, \mathbf{x}_1)$ as a c-VAE, we define a decoder network $G_{\theta_{\text{mjr}}}(\mathbf{z}, \mathbf{x}_1)$ that parameterizes the likelihood $p_{\theta_{\text{mjr}}}(\mathbf{y}|\mathbf{z}, \mathbf{x_1})$, and its corresponding encoder network $E_{\phi_{\text{mjr}}}(\mathbf{y}, \mathbf{x}_1)$ that parame-

terizes the approximate posterior $q_{\phi_{\mathrm{mjr}}}(\mathbf{z}|\mathbf{y},\mathbf{x_1})$, both conditioned on attribute labels $\mathbf{x_1}$. They are trained on majority samples by maximizing the standard ELBO loss. A function $\mathbf{x}_2 = \hat{m}(\mathbf{x}_1)$ is estimated on attribute values from majority samples.

**Stage II:** To leverage the learned $g_{\mathrm{mjr}}(\mathbf{z}, \mathbf{x}_1)$, we now expand the decoder $G_{\theta_{\mathrm{mjr}}}$ with several additional layers before the final activation layer $\sigma$, which we denote as $G_{\theta_{\mathrm{mjr}}\backslash\sigma}$. We denote this expanded portion of the decoder as $G_{\theta_{\mathrm{res}}}(G_{\theta_{\mathrm{mjr}}\backslash\sigma}, \mathbf{x}_1, \gamma)$, conditioned on $\gamma = \mathbf{x}_2 - \hat{m}(\mathbf{x}_1)$. As illustrated in Fig. 6A, we expand the encoder network with additional layers $E_{\phi_{\mathrm{res}}}$ in a mirror of the expanded decoder network as $E_{\phi_{\mathrm{res}}}(\mathbf{y}_{\mathrm{mnr}}, \gamma)$, to produce output that will serve as the input to the first-stage encoder. The expanded networks are trained on minority samples by maximizing the ELBO loss below, where we keep the stage-I weights $\theta_{\mathrm{mjr}}$ and $\phi_{\mathrm{mjr}}$ frozen:

$$\max_{\theta_{\mathrm{res}},\phi_{\mathrm{res}}} \left\{ \mathbb{E}_{\mathbf{z}\sim E_{\phi_{\mathrm{mjr}}}(E_{\phi_{\mathrm{res}}}(\mathbf{y}_{\mathrm{mnr}},\gamma),\mathbf{x}_1)} \left[ \|\mathbf{y}_{\mathrm{mnr}} - \hat{\mathbf{y}}_{\mathrm{mnr}}\|_2^2 \right] + \beta \underbrace{D_{\mathrm{KL}}\big(E_{\phi_{\mathrm{mjr}}}(E_{\phi_{\mathrm{res}}}(\mathbf{y}_{\mathrm{mnr}},\gamma),\mathbf{x}_1) \,\|\, p(\mathbf{z})\big)}_{\text{KL Divergence}} \right\} \quad (3)$$

$$\text{where} \quad \hat{\mathbf{y}}_{\mathrm{mnr}} = G_{\theta_{\mathrm{res}}}(G_{\theta_{\mathrm{mjr}}\backslash\sigma}(\mathbf{z},\mathbf{x}_1),\mathbf{x}_1,\gamma), \quad \mathbf{z} \sim E_{\phi_{\mathrm{mjr}}}(E_{\phi_{\mathrm{res}}}(\mathbf{y}_{\mathrm{mnr}},\gamma),\mathbf{x}_1) \quad (4)$$

While Equation (4) represents a general formulation for residual generation, on simpler datasets, an additive residual can be considered such that $\hat{\mathbf{y}}_{\mathrm{mnr}} = G_{\theta_{\mathrm{mjr}}}(\mathbf{z},\mathbf{x_1}) + G_{\theta_{\mathrm{res}}}(G_{\theta_{\mathrm{mjr}}\backslash h}(\mathbf{z},\mathbf{x}_1)),\gamma)$. With Equation (4), $G_{\theta_{\mathrm{res}}}$ leverages the feature map of the stage-I generator $G_{\theta_{\mathrm{mjr}}}$ and the residual $\gamma$ to resolve the difference between majority and minority generations. Intuitively, because $G_{\theta_{\mathrm{mjr}}}$ and $E_{\phi_{\mathrm{mjr}}}$ are trained in stage-I to generate and encode from majority samples, the expanded $G_{\theta_{\mathrm{res}}}$ will be encouraged to learn to modify a majority-image feature map to include features corresponding to the residual partialled-out from $\mathbf{x}_2$, while the expanded $E_{\theta_{\mathrm{res}}}$ will be encouraged to alter such features to generate an output feature map acceptable to $E_{\phi_{\mathrm{mjr}}}$ (*i.e.,* compliant with feature map seen by $E_{\phi_{\mathrm{mjr}}}$ in stage-I). With this, we shift the burden of learning the c-VAE mainly to majority samples, and allow the use of limited minority samples for learning the necessary reisdual changes only.

## 4.3 PROREGEN-GAN

**Stage I:** To learn $g_{\mathrm{mjr}}(\mathbf{z}, \mathbf{x}_1)$ as a c-GAN, we define a generator $G_{\theta_{\mathrm{mjr}}}(\mathbf{z}, \mathbf{x}_1)$ and discriminator $D_{\phi_{\mathrm{mjr}}}(\mathbf{y}, \mathbf{x}_1)$, both conditioned on attribute $\mathbf{x}_1$. They are trained on majority samples using standard adversarial loss. A function $\mathbf{x}_2 = \hat{m}(\mathbf{x}_1)$ is estimated on attribute values from majority samples.

**Stage II:** Similar to the setting of c-VAE, we now expand the generator $G_{\theta_{\mathrm{mjr}}}$ with several additional layers, denoted as $G_{\theta_{\mathrm{res}}}$, starting with the feature map produced by $G_{\theta_{\mathrm{mjr}}}$ before the final activation layer. As illustrated in Fig. 6B, we expand the discriminator with additional layers $D_{\phi_{\mathrm{res}}}$ in a mirrored fashion. Both $G_{\theta_{\mathrm{res}}}$ and $D_{\theta_{\mathrm{res}}}$ are conditioned on $\gamma = \mathbf{x}_2 - \hat{m}(\mathbf{x}_1)$. The expanded networks are trained on minority samples using the adversarial loss while freezing stage-I network weights $\theta_{\mathrm{mjr}}$ and $\phi_{\mathrm{mjr}}$:

$$\min_{\theta_{\mathrm{res}}} \max_{\phi_{\mathrm{res}}} \left\{ \mathbb{E}_{\mathbf{y}_{\mathrm{mjr}},\mathbf{x}_1,\gamma\sim p_{\mathrm{data}}} \left[ \log D_{\phi_{\mathrm{mjr}}}(D_{\phi_{\mathrm{res}}}(\mathbf{y}_{\mathrm{mnr}},\gamma),\mathbf{x}_1) \right] \right.$$

$$\left. + \mathbb{E}_{\mathbf{z}\sim p(\mathbf{z}),\mathbf{x}_1,\gamma\sim p_{\mathrm{data}}} \left[ \log\big(1 - D_{\phi_{\mathrm{mjr}}}\big(D_{\phi_{\mathrm{res}}}(\hat{\mathbf{y}}_{\mathrm{mnr}},\gamma),\mathbf{x}_1\big)\big) \right] \right\} \quad (5)$$

$$\text{where} \quad \hat{\mathbf{y}}_{\mathrm{mnr}} = G_{\theta_{\mathrm{res}}}(G_{\theta_{\mathrm{mjr}}\backslash\sigma}(\mathbf{z},\mathbf{x}_1),\mathbf{x}_1,\gamma), \quad \mathbf{z} \sim \mathcal{N}(\mathbf{0},\mathbf{I}) \quad (6)$$

Similarly, with Equation (6), the expanded $G_{\theta_{\mathrm{res}}}$ learns to use the residual attribute $\gamma$ to change the distribution of generated majority samples to one that aligns with the distribution of real minority images $\mathbf{y}_{\mathrm{mnr}}$. At the same time, the expanded discriminator $D_{\phi_{\mathrm{res}}}$ is encouraged to change the residual features on the real/generated minority sample in order to leverage the stage-I discriminator $D_{\phi_{\mathrm{mjr}}}$ that has learned to work with the distribution of majority samples.

One difference between Equation (4) for ProReGen-VAE and Equation (6) for ProReGen-GAN is the distribution over which the sample $\mathbf{z}$ is taken. This is inherently determined by the training loss of the two models, where the likelihood loss of VAE is calculated over the posterior distribution of $\mathbf{z}$ conditioned on an observed image (emphasizing instance-level reconstruction) *vs.* the adversarial loss in GAN is calculated over the prior density of $\mathbf{z}$ (emphasizing distribution-level distance).

## 4.4 PROREGEN-DM

**Stage I:** We follow the standard denoising diffusion probabilistic model (DDPM) (Ho et al., 2020) to learn c-DM that capture attribute correlation. We noise the majority samples $\mathbf{y}_{\mathrm{mjr}_0} \in \mathbb{R}^{C \times H \times W}$

using the closed form noising expression in Equation (7) derived from the forward noising process $q(\mathbf{y}_{\mathrm{mjr}_t} \mid \mathbf{y}_{\mathrm{mjr}_{t-1}}) := \mathcal{N}(\mathbf{y}_{\mathrm{mjr}_t}; \sqrt{1 - \beta_t}\,\mathbf{y}_{\mathrm{mjr}_{t-1}}, \beta_t I)$, where $\{\beta_t\}_{t=1}^{T}$ is the noise schedule. To obtain $g_{\mathrm{mjr}}(\mathbf{z}, \mathbf{x}_1)$, we learn the reverse denoising process parameterized by $\epsilon_{\theta_{\mathrm{mjr}}}(\mathbf{y}_{\mathrm{mjr}_t}, t, \mathbf{x}_1)$, conditioned on attribute label $\mathbf{x}_1$, using the loss function in Equation (8).

$$\mathbf{y}_t = \sqrt{\bar{\alpha}_t}\,\mathbf{y}_0 + \sqrt{1 - \bar{\alpha}_t}\,\epsilon, \qquad \epsilon \sim \mathcal{N}(0, I), \qquad \bar{\alpha}_t = \prod_{s=1}^{t} \alpha_s, \; \alpha_t = 1 - \beta_t \qquad (7)$$

$$\mathcal{L}_{\mathrm{MSE}}(\theta) = \mathbb{E}_{\mathbf{y}_0, t, \mathbf{c}, \epsilon}\big[\|\epsilon - \epsilon_\theta(\mathbf{y}_t, t, \mathbf{c})\|^2\big] \qquad \text{where} \quad \mathbf{c} = \mathbf{x}_1 \text{ (Stage-I) or } \gamma \text{ (Stage-II)} \qquad (8)$$

**Stage II:** Given the unique design of DM, we cannot simply expand the learned denoising process in stage-I with additional denoising steps. Instead, as illustrated in Fig. 6C, we introduce a trainable stage-II network $\epsilon_{\theta_{\mathrm{mnr}}}(\mathbf{y}_{\mathrm{mnr}_t}, t, \gamma)$ conditioned on the residual attribute $\gamma$, trained with only minority samples using Equations (7-8), but injected with features learned by the stage-I $\epsilon_{\theta_{\mathrm{mjr}}}(\mathbf{y}_{\mathrm{mjr}_t}, t, \mathbf{x}_1)$ – this is inspired by the design of control-net (Zhang et al., 2023), but distinct in that we use the *fronzen stage-I network* to provide features for the stage-II network. We consider $\mathbf{y}_{\mathrm{mjr}_t}$ input of $\epsilon_{\theta_{\mathrm{mjr}}}$ as a corresponding majority version of $\mathbf{y}_{\mathrm{mnr}_t}$ with all aspects except the residual feature identical. We inject stage-I features in the downsampling and middle blocks of U-Net (Ronneberger et al., 2015), but not in the upsampling blocks as we empirically observed better quality without it. We use the same variance schedule, diffusion steps, and architecture design in stage-II network so that the features coming from stage-I are aligned to the particular diffusion step of stage-II being processed.

## 5 EXPERIMENTS AND RESULTS

**Data:** For quantitative evaluations that require an oracle classifier trained on balanced attributes, we consider Colored-MNIST (Lee et al., 2021), MNIST-Correlation (Mu & Gilmer, 2019), and Corrupted-CIFAR10 (Hendrycks & Dietterich, 2019). We assume known labels of the attributes that are correlated. For the two MNIST datasets, we curated high levels of correlation strengths at 95%, 98%, 99%, and 99.5%, where the % represents the percentage of majority training samples in the training data. For Corrupted-CIFAR10 derived from natural images, we considered a wider range of correlation strengths of 70%, 80%, 90%, and 99%. For each dataset, we included a *balanced* version without any attribute correlations to both establish *oracle* attribute classifiers and establish a reference performance for all models considered. To test on datasets with natural attribute correlations, we considered CelebA for qualitative evaluation (due to the lack of an oracle classifier). Details of the dataset are described in their respective sections below. Test accuracies of their oracle classifiers are in Appendix B and their training data distribution in Appendix C.

**Baselines:** We considered c-VAE, c-GAN, and c-DM baselines with the following strategies for mitigating attribute correlations: 1) naive, 2) re-weighting, achieved with upsampling minority samples using the *weighted random sampler* and 3) pseudo-supervision on counterfactual generations, represented by causal-cHVAE (Ribeiro et al., 2023) where a classifier is used to finetune the model in an optional second-stage of counterfactual generation, and causal-GAN (Kocaoglu et al., 2017) where the attribute classifier is incorporated in the end-to-end adversarial training. We could not identify causal formulations of DMs that accommodate multiple correlated attributes. We present the architectural details and parameter counts of all baselines and ProReGen in Appendix D.

**Evaluation:** We evaluated the performance of all DGMs in generating both majority and minority samples. To generate with the trained DGMs, we sampled from $\mathbf{z} \sim \mathcal{N}(\mathbf{0}, \mathbf{I})$ and generated a total of 25,000 samples, with equal number of samples for each unique attribute combination, per dataset. We evaluated generated samples using: 1) correctness, measuring the ratio of generations in which the attributes evaluated by the oracle classifier match the intended attributes; 2) Fréchet Inception Distance (FID) (Heusel et al., 2017), measuring the quality and diversity of generations by comparing the representations (retrieved from *InceptionV3* network) of the generated samples against a test set of diverse real samples; and 3) coverage & density (Naeem et al., 2020), measuring the diversity and fidelity, respectively, of generations compared with a test set of diverse real samples.

### 5.1 EXPERIMENTS & RESULTS ON COLORED-MNIST

**Settings of Attribute Correlations:** Colored-CMNIST (Lee et al., 2021) is a commonly-used benchmark for synthesizing attribute correlations in the training data. It is an MNIST-variant with a

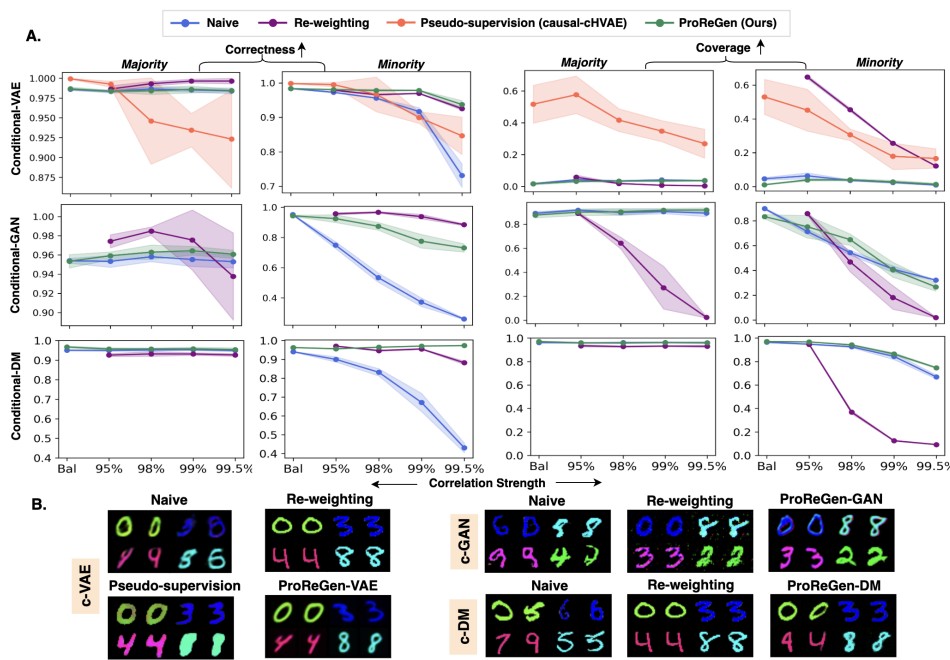

Figure 2: A: Correctness and Coverage of majority and minority generations from ProReGen *vs.* baselines for Colored-MNIST. B: Visual examples of minority generations at 99.5% correlation.

distinct majority color for each of the 10 digits. This creates a correlation between *digit* and *color* attributes, both discrete but non-binary. All baselines as described were considered in this dataset, with the exception of causal-GAN which was designed to work with binary labels (Kocaoglu et al., 2017). We considered *digit* $\rightarrow$ *color* as the causal direction $x_1 \rightarrow x_2$ for our experiments.

**Results and Analysis:** We present some representative quantitative metrics and visual examples on Colored-MNIST in Fig. 2. Complete results are included in Appendix K.1. As shown, ProReGen (green) in general improved the correctness of minority generations in comparison to the naively trained baseline (blue, significantly more in c-GAN and c-DM), at comparable quality metrics. In comparison, while causal-cHVAE (red, the baseline that leverages pseudo-supervision) also improved the correctness of minority generations, this improvement was obtained at the expense of degraded correctness in majority generations, suggesting that the use of pseudo-supervision has introduced trade-off in the correctness of majority *vs.* minority generations at higher correlation strengths. ProReGen, in comparison, was relatively stable across correlation strengths for majority generations. Note that the relatively strong quality metrics of causal-cHAVE may be due to its base hierarchical VAE architecture that was different from the rest of the VAE models.

Re-weighting resulted in comparable correctness metrics in both majority and minority generations in comparison to ProReGen. However, in all models, signs of overfitting could be observed in minority generations hence limited diversity, as shown in the rapidly-degrading Coverage metrics as the correlation strength increased in Fig. 2A and the lack of diversity in the visual examples in Fig. 2B. In re-weighted c-VAE, the higher coverage of minority generations in comparison to ProReGen can seem counterintuitive. A potential reason was that generations from learned distribution in ProReGen were comparatively more blurred compared to memorization of training samples.

## 5.2 EXPERIMENTS & RESULTS ON MNIST-CORRELATION

**Settings of Attribute Correlations:** MNIST-Correlation (Mu & Gilmer, 2019) is another MNIST variant where most of the even digits are clean and most of the odd digits include zigzag, hence resulting in a correlation between attributes $\mathbf{x_1} = \{\text{even, odd}\}$ and $\mathbf{x_2} = \{\text{clean, zigzag}\}$. We similarly created correlation strengths at 95%, 98%, 99%, and 99.5% following (Goel et al., 2020). Along with the information on *presence / absence* of zigzag, we also added the coordinates of

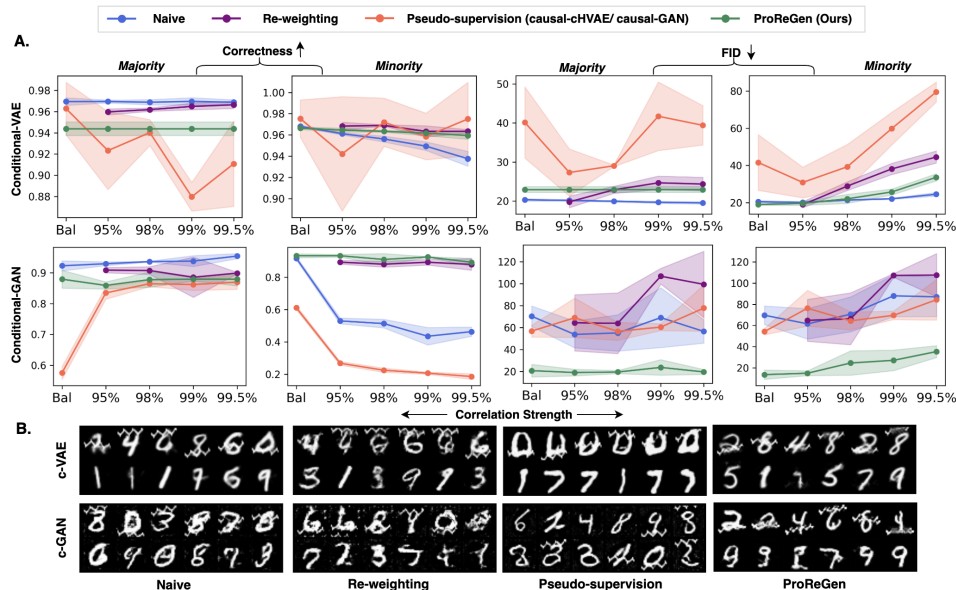

Figure 3: A: Correctness and FID of majority and minority generations from ProReGen *vs.* baselines for MNIST-Correlation. B: Visual comparison of minority generations at 99.5% correlation ratio. In each image grid, the intended generation is zigzag-even on the top, and clean-odd for the bottom.

the end points of zigzag (mid-point of image in case of *clean* image) as additional feature attributes in $x_2$ to represent residual attributes that cannot be predicted from $x_1$ but will contribute to the generation of images. Additional details on this are included in Appendix D. We consider *even / odd → presence / absence of zigzag* as the causal direction for our experiments.

**Results and Analysis:** We present quantitative results and visual examples for MNIST-Correlation in Fig. 3, with complete results in Appendix K.2. c-DMs were not included because the highest level of correlation strengths did not induce a bias in its naive version. Compared to naive c-VAE (blue), ProReGen-VAE (green) improved the correctness of minority generations, at some degradation in the correctness of majority generations and similar or slightly worsened quality metrics. In comparison, causal-cHVAE (red) was inconsistent in improving the correctness of minority generation at more significance deterioration of both majority correctness and FID metrics. Re-weighting (purple) delivered similar correctness in minority generations and quality metrics in majority generations, but better majority correctness and worsened FID (reflecting diversity issue) in minority generations.

Compared to naive c-GAN (blue), ProReGen-GAN (green) significantly improved the correctness of minority generations along with significantly improved FID in both generations with moderate degradation of correctness in majority generations. Causal-GAN (red) was not successful in improving the correctness of minority generations, with FID metrics similar to the naive baseline. Re-weighting (purple) improved correctness of minority generations with slight compromise in the correctness of majority generations, but also worsened FID metrics.

## 5.3 EXPERIMENTS & RESULTS ON CORRUPTED-CIFAR10

**Settings of Attribute Correlations:** We adopted CIFAR10 (Krizhevsky et al., 2009) and curated it following the practice in (Hendrycks & Dietterich, 2019) to create a correlation between object classes and image corruption types. More specifically, we considered five different object classes, $x_1 = \{$car, bird, dog, horse, ship$\}$, and applied a unique type of corruption, $x_2 = \{$gaussian noise, shot noise, impulse noise, contrast, brightness$\}$, respectively, to the majority of the training samples per object class. The minority samples have remaining corruptions uniformly sampled at random. We considered correlation strengths at the level of 70% and 80% for GAN-based models and 90% and 99% for DM-based models because they were the levels at which bias was in-

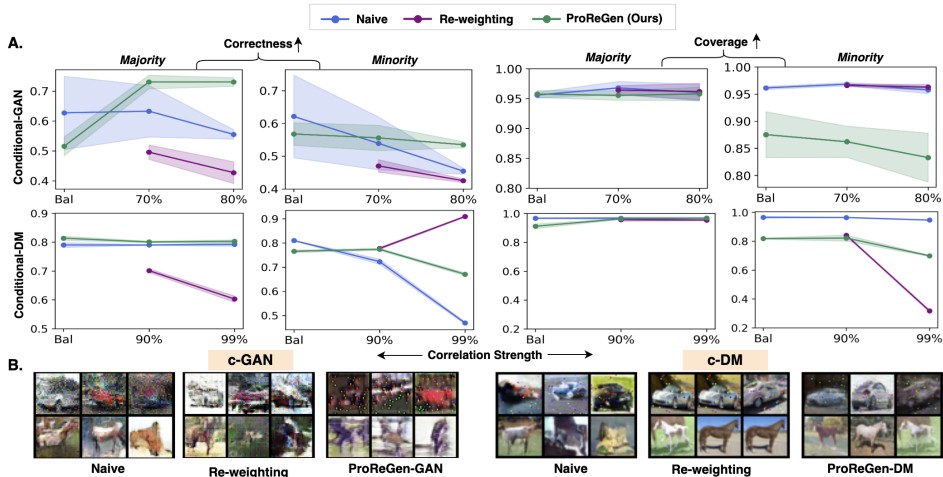

Figure 4: A: Correctness and Coverage of generations on Corrupted-CIFAR10. B: Visual examples of minority generations at 80% correlation for c-GAN and 99% correlation for c-DM. In each image grid, the intended generation is: top – {car, impulse noise}; bottom – {horse, brightness}.

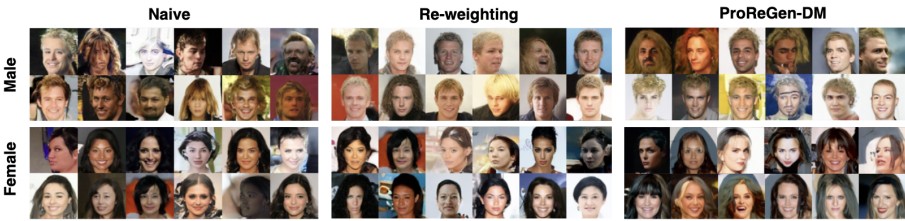

Figure 5: Visual examples of minority generations on CelebA under natural correlations.

duced in the respective naive versions of these models. We did not test VAE-based models due to their quality issues in generation (blurring of corruption details) on these natural images.

**Results and Analysis:** Representative results in Fig. 4 showed that both naive c-GAN and c-DM (blue) suffered from increasing correlation strengths, especially in the correctness of minority generation. Re-weighting (purple) sometimes addressed this issue (in re-weighted c-DM), but at the expense of decreasing diversity (c-DM) and deteriorated majority correctness (c-DM and c-GAN). In comparison, ProReGen (green) improved the correctness of minority generations, without compromising majority generations and with limited degradation of image qualities.

## 5.4 EXPERIMENTS & RESULTS ON CELEBA

**Settings of Attribute Correlations:** We considered multi-attribute face dataset CelebA (Liu et al., 2015) and the natural correlation between $x_1$ = Gender and $x_2$ = Hair Color; most blond-haired individuals are Females (22.9K) with only around 1.3K blond-haired males in the entire training dataset. We considered four subgroups (Male/Female, Blond/Black Hair) with the naturally existing correlation in the dataset. Additionally, we experimented with reducing the count of {Female-Black Hair} subgroup to be close to that of {Male-Blond Hair}. We tested only DM-based models here given their stronger performance in previous experiments.

**Results and Analysis:** Fig. 5 provide examples of minority generations for models trained under natural correlation, with additional results provided in Appendix K.4. Visually, naive c-DMs suffered evidently in the generation of minority images both in attribute correctness and generation quality. ProReGen-DM was able to consistently generate realistic minority images with correct attribute combinations. While it is difficult to pinpoint its performance over re-weighting without further quantitative analysis (due to the lack of a good oracle classifier), these results demonstrated the utility of ProReGen in realistic image datasets with natural attribute correlations.

Table 1: A comparison of progressive two-stage training *vs.* simultaneous training of $g_{\mathrm{mjr}}$ and $g_{\mathrm{res}}$ in ProReGen-GAN for 95% correlation strength in Colored-MNIST

|  |  | Correctness ↑ | FID ↓ | Coverage ↑ | Density ↑ |
|---|---|---|---|---|---|
| **Two-Staged Training** | **Majority** | $0.9592 \pm 0.004$ | $16.9488 \pm 2.3446$ | $0.9003 \pm 0.0275$ | $0.7628 \pm 0.0386$ |
|  | **Minority** | $0.9256 \pm 0.0257$ | $17.2562 \pm 7.8816$ | $0.7519 \pm 0.0903$ | $0.6089 \pm 0.1216$ |
| **Single-Staged Training** | **Majority** | $0.9289 \pm 0.0369$ | $29.0843 \pm 6.3397$ | $0.7636 \pm 0.0216$ | $0.5585 \pm 0.0114$ |
|  | **Minority** | $0.3557 \pm 0.1689$ | $65.0227 \pm 12.3787$ | $0.0432 \pm 0.0335$ | $0.0320 \pm 0.0220$ |

## 5.5 Additional Ablation Studies

**Effects of Progressive training:** To demonstrate the benefit of the progressive two-stage training, we performed an ablation of ProReGeN where the model architecture remained the same but $g_{\mathrm{mjr}}$ and $g_{\mathrm{res}}$ were optimized simultaneous *vs.* progressively in two stages. As shown in Table 1, without progressive training, there was minimal to no effect on the correctness of majority generations but some impact on its quality (*e.g.,* a drop of 27% in density). The generation of minority samples however was significantly worsened (*e.g.,* a drop of 62% in correctness and nearly three times worse in FID). We further demonstrate this with sample minority generations in Fig. 7 in Appendix.

**Sensitivity to errors in $m(\mathbf{x}_1)$:** To understand the sensitivity of ProReGen to errors in the estimation of $m(\mathbf{x}_1)$, we perturbed $\mathbf{x}_1$ to $\hat{\mathbf{x}}_1$ via uniform random shift, within the valid domain, to simulate a wrong estimation of $m(\mathbf{x}_1)$. We experimented with three levels of perturbation with increasing percentage of samples induced with random shifting in attribute $\mathbf{x}_1$ per training epoch. We present our observation in Table 12 in Appendix G for ProReGen-GAN trained on Colored-MNIST with 95% correlation strength. The correctness of ProReGen-GAN dropped as expected with the increase in the level of perturbation, although not rapidly and still improved over the naive model at $80\%$ of errors. A closer inspection showed that the drop resulted from inaccuracy in color generation, which is as expected since $\gamma$ dictates the color residual between majority and minority samples.

**Computation cost and additional training details:** We present the computation cost, convergence plots, and a small analysis of sensitivity to residual sub-networks in Appendix H – J.

**Effect of assumed causal directions between attributes:** We examined the effect of the assumption of causal directions between attributes by inverting the causal direction *digit* $\rightarrow$ *color* to *color* $\rightarrow$ *digit* for Colored-MNIST. We considered ProReGen-GAN for our analysis. We observed that the performance with the inverted causal direction *color* $\rightarrow$ *digit* was suboptimal, with only $0.0811 \pm 0.0127$ correctness, on average, of minority generations *vs.* $0.9396 \pm 0.0016$, on average, with *digit* $\rightarrow$ *color*. The correctness of majority generations was similar. This indicated that learning the residual for digit conversion was much more difficult. We present the generation samples along with additional results in Appendix F. This suggests that the difficulty, and hence performance of residual generation task, is influenced by the causal direction assumed and should be used to design the attribute causal direction for ProReGen in practice (unless the true causal direction is known).

## 6 Conclusions & Discussions

We present ProReGen, a novel DGM-design that employs progressive training and leverages majority training samples to learn most part of the generation task while employing minority training samples to only learn the residual information. We demonstrate its benefit in improving generation correctness against the baselines using synthetic and natural images at different correlation ratios. **Limitation:** ProReGen-VAE and -GAN perform residual operation at the image level. While ProReGen-DM leveraged feature injection, the base DDPM considered performs diffusion at the pixel space. Future works will investigate extending the concept of ProReGen to realize the effect of residual $\gamma$ on image generation through the latent space. ProReGen as presented assumes the ability to separate samples into discrete subgroups; future work will extend it to settings with continuous attributes and use attribute residual to modulate sample importance in stage I *vs.* II learning instead. Evaluation of minority generation remains a significant challenge: since it is not possible to have a perfect oracle classifier due to reasons such as natural attribute imbalance, future works should assess the uncertainty of these correctness metrics. Finally, the challenge of attribute correlation can be expected to persist and even amplify in text-to-image models due to the sparsity and heavy-tailed nature of the problem space, leaving an exciting avenue for future explorations.

## REPRODUCIBILITY STATEMENT

We present the design of network architectures and training details used in our proposed method in Appendix D. Moreover, we provide references to the official code repositories employed for experimentation with two of our baselines in the same section. The source code for ProReGen is available at `https://github.com/ruby-stha/ProReGen_ICLR2026`. Additionally, we share the training data distribution of the datasets used to present our results in Appendix C.

## ACKNOWLEDGEMENTS

This study is supported by National Institutes of Health / National Heart, Lung, and Blood Institute (NIH/NHLBI) Grant R01HL145590.

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

# A  ILLUSTRATION OF PROREGEN

Fig. 6 illustrates the outline of the instantiations of ProReGen on c-VAE (A), c-GAN (B), and c-DM (C).

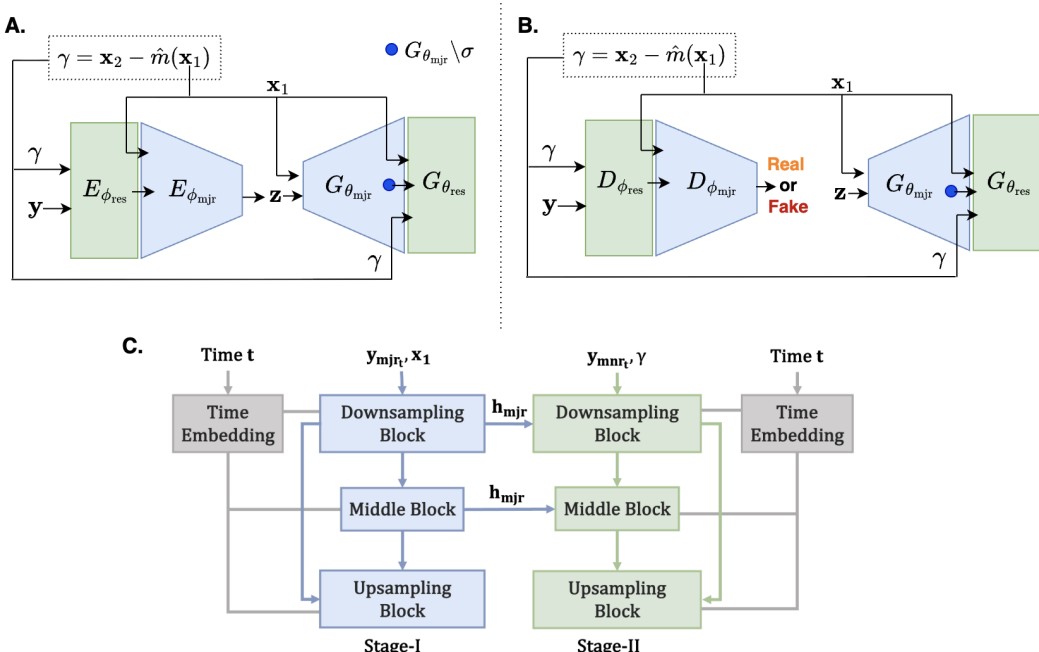

Figure 6: Illustration of ProReGen-VAE (A), ProReGen-GAN (B), and ProReGen-DM (C) .

## B  ORACLE PERFORMANCE FOR EACH DATASET

We present the performance oracle classifiers for Colored-MNIST, MNIST-Correlation, and Corrupted-CIFAR10 in Table 2.

Table 2: Test Accuracy of Oracle Classifiers for Colored-MNIST, MNIST-Correlation, and Corrupted-CIFAR10

|  | **Oracle Classifier For** | **Average Test Accuracy** |
|---|---|---|
| **Colored-MNIST** | Digit | 0.958 |
|  | Color | 1.0 |
| **MNIST-Correlation** | Even/Odd Digit Type | 0.965 |
|  | Presence/Absence of Zigzag | 1.0 |
| **Corrupted-CIFAR10** | Object Type | 0.849 |
|  | Corruption Type | 0.99 |

## C  TRAINING DATA DISTRIBUTION FOR EACH DATASET

We present the count of majority and minority training samples employed across each correlation ratio for Colored-MNIST in Table 3, for MNIST-Correlation in Table 6, and for Corrupted-CIFAR10 in Table 5.

Table 3: Counts of Majority and Minority Samples Across Varying Levels of Correlation Strengths Explored for Colored-MNIST. For the *Balanced* setting, we consider equal number of samples per (digit, color) combination, *i.e.,* around 550 samples per combination. The total number of unique combinations is 100.

| **Correlation Strength** | **Minority** | **Majority** |
|---|---|---|
| 95% | 2450 | 52552 |
| 98% | 986 | 54014 |
| 99% | 492 | 54510 |
| 99.5% | 249 | 54751 |

Table 4: Counts of Majority and Minority Samples Across Varying Levels of Correlation Strengths Explored for MNIST-Correlation. For the *Balanced* setting, we consider equal number of samples per attribute combination, *i.e.,* around 20,000 samples per combination. The total number of unique combinations is 4.

| Correlation Strength | Minority | Majority |
|---|---|---|
| 95% | 2104 | 40000 |
| 98% | 816 | 40000 |
| 99% | 404 | 40000 |
| 99.5% | 200 | 40000 |

Table 5: Counts of Majority and Minority Samples Across Varying Levels of Correlation Strengths Explored for Corrupted-CIFAR10. For the *Balanced* setting, we consider equal number of samples per attribute combination, *i.e.,* around 800 samples per combination. The total number of unique combinations is 25.

| Correlation Strength | Minority | Majority |
|---|---|---|
| 70% | 6000 | 14000 |
| 80% | 4000 | 16000 |
| 90% | 2000 | 18000 |
| 99% | 200 | 19800 |

Table 6: Counts of Majority and Minority Samples in natural and 94% Correlation Strengths Explored for CelebA. The counts for Majority and Minority are such that (Male, Blond Hair) and (Female, Black Hair) are considered in Minority, while (Male, Black Hair) and (Female, Blond Hair) in Majority.

| Correlation Strength | Minority | Majority |
|---|---|---|
| Natural | 20170 | 43001 |
| 94% | 2774 | 43001 |

## D  IMPLEMENTATION DETAILS

**Network Architecture.** We employ a stack of convolution and transposed convolution layers to implement the c-VAE (Sohn et al., 2015) and follow DCGAN-like architectural setup for c-GAN (Radford et al., 2015). For causal-cHVAE and causal-GAN, we follow their official code repositories: Ribeiro et al. (2023) for causal-cHVAE and Kocaoglu et al. (2017) for causal-GAN.

In both c-VAE and c-GAN, since the residual effect generator only requires adjusting residual features on images for which most of the generative factors have already been produced by the majority DGM, the expanded layers are lightweight, comprising relatively low parameter count than the majority DGMs (Table 11). Moreover, they are also designed such that they maintain the spatial dimension of the input image (*e.g.,* using convolutional operations with $kernel = 3$, $stride = 1$, $padding = 1$). We present the details of their network architecture in Table 7-10.

For c-DM, we use the implementation from Nichol & Dhariwal (2021) for the *naive* baseline and extend it to support the *re-weighting* baseline. Building on the foundation from Nichol & Dhariwal (2021), we further adapt its code to incorporate our model design (shown in Fig. 6C), including feature injection and required conditioning. We consider condition integration via channel-wise concatenation in ProReGen-DM and maintain consistent conditioning method across all its baselines.

We present a summary of the parameter counts for the stage-I and stage-II models of ProReGen for all considered datasets in Table 11.

**Conditional Information** $\gamma$**.** Residual (orthogonal) attribute $\gamma = x_2 - m(x_1)$ is broadcasted to match spatial size of the input and concatenated as additional channels to provide it as conditional

information to the expanded layers. When we have high-dimensional attribute $x_2$, where only part of it is predictable from $x_1$, we predict the predictable dimensions by $m(x_1)$ to get $\gamma$. As *implementation choice*, we can then either keep the additional (unpredictable) dimensions as additional channels or use them to manipulate (*e.g.,* mask) predictable dimensions. We employ the latter approach for MNIST-Correlation for ProReGen, where we mask the $\gamma$ information using the information of line joining the two end-points of zigzag. We employ the same approach in the encoder of naively-trained conditional-VAE and its re-weighted version. However, employing such masking in naively-trained conditional-GAN led to the model ignoring the even/odd label information, potentially due to the discriminator relying on the now easier feature, zigzag (due to its masking). Therefore, we appended the coordinate information as additional channels during conditioning for conditional-GAN.

Table 7: The architecture of the expanded portion of generator network for Colored-MNIST. The architecture of the expanded portion of discriminator or encoder is a mirror of it. Here $C_{img}$ denotes the number of image channels and $d_t$ is the dimension of $\gamma$.

| Part | Output Shape | Layer Information |
|------|-------------|-------------------|
| Input | (B, $C_{\text{img}} + d_t$, H, W) | – |
| Conv Block 1 | (B, 64, H, W) | Conv2d($C_{\text{img}} + d_t$, 64, 3, 1, 1), GroupNorm(8, 64), ReLU |
| Conv Block 2 | (B, 32, H, W) | Conv2d(64, 32, 3, 1, 1), GroupNorm(8, 32), ReLU |
| Output Layer | (B, $C_{\text{img}}$, H, W) | Conv2d(32, $C_{\text{img}}$, 3, 1, 1) |

Table 8: The architecture of the expanded portion of decoder network for MNIST-Correlation in ProReGen-VAE. The architecture of the expanded portion of encoder is a mirror of it. Here $C_{img}$ denotes the number of image channels and $d_t$ is the dimension of $\gamma$.

| Part | Output Shape | Layer Information |
|------|-------------|-------------------|
| Input | (B, $C_{\text{img}} + d_t$, H, W) | – |
| Conv Block 1 | (B, 64, H, W) | Conv2d($C_{\text{img}} + d_t$, 64, 3, 1, 1) |
| Conv Block 2 | (B, 64, H, W) | Conv2d(64, 64, 3, 1, 1), ReLU, Conv2d(64, 64, 3, 1, 1) |
| Conv Block 3 | (B, 64, H, W) | Conv2d(64, 64, 3, 1, 1), ReLU, Conv2d(64, 64, 3, 1, 1) |
| Output Layer | (B, $C_{\text{img}}$, H, W) | Conv2d(64, $C_{\text{img}}$, 3, 1, 1) |

Table 9: The architecture of the expanded portion of generator network for MNIST-Correlation in ProReGen-GAN . The architecture of the expanded portion of discriminator is a mirror of it. Here $C_{img}$ denotes the number of image channels and $d_t$ is the dimension of $\gamma$. This is similar to the architecture employed in ProReGen-VAE, shown in Table D, with a light extension to the design and the addition of normalization layers to enhance the training stability.

| Part | Output Shape | Layer Information |
|------|-------------|-------------------|
| Input | (B, $C_{\text{img}} + d_t$, H, W) | – |
| Conv Block 1 | (B, 64, H, W) | Conv2d($C_{\text{img}} + d_t$, 64, 3, 1, 1), GroupNorm(8, 64), ReLU |
| Conv Block 2 | (B, 64, H, W) | Conv2d(64, 64, 3, 1, 1), GroupNorm(8, 64) |
| Conv Block 3 | (B, 32, H, W) | Conv2d(64, 32, 3, 1, 1), GroupNorm(8, 32), ReLU |
| Conv Block 4 | (B, 32, H, W) | Conv2d(32, 32, 3, 1, 1), GroupNorm(8, 32) |
| Output Layer | (B, $C_{\text{img}}$, H, W) | Conv2d(32, $C_{\text{img}}$, 3, 1, 1) |

Table 10: The architecture of the expanded portion of generator network for Corrupted-CIFAR10. The architecture of the expanded portion of discriminator is a mirror of it. Here, $C_{img}$ denotes the number of image channels and $d_t$ denotes the dimension of $\gamma$ and $\mathbf{x}_1$.

| Part | Output Shape | Layer Information |
|---|---|---|
| Residual Block 1 | (B, $C_{\text{mid}}$, H, W) | Concatenate $h_{\text{mjr}}, \gamma, \mathbf{x}_1$ along channel dim
Conv2d($C_{\text{img}} + 2d_t$, $C_{\text{mid}}$, 3, 1, 1), GroupNorm(8, $C_{\text{mid}}$), ReLU
Conv2d($C_{\text{mid}}$, $C_{\text{mid}}$, 3, 1, 1), GroupNorm(8, $C_{\text{mid}}$)
Skip connection: Conv2d($C_{\text{img}}$, $C_{\text{mid}}$, 1, 1, 0) (or Identity if channels match)
Element-wise addition (residual connection) |
| Residual Block 2 | (B, $C_{\text{mid2}}$, H, W) | Concatenate Residual Block 1 output, $\gamma, \mathbf{x}_1$ along channel dim
Conv2d($C_{\text{mid}} + 2d_t$, $C_{\text{mid2}}$, 3, 1, 1), GroupNorm(8, $C_{\text{mid2}}$), ReLU
Conv2d($C_{\text{mid2}}$, $C_{\text{mid2}}$, 3, 1, 1), GroupNorm(8, $C_{\text{mid2}}$)
Skip connection: Conv2d($C_{\text{mid}}$, $C_{\text{mid2}}$, 1, 1, 0) (or Identity if channels match)
Element-wise addition (residual connection) |
| Output Layer | (B, $C_{\text{img}}$, H, W) | Conv2d($C_{\text{mid2}}$, $C_{\text{img}}$, 3, 1, 1) |

Table 11: Parameter counts of Stage-I and Stage-II of ProReGen for different models and datasets

| Datasets | Models | Stage-I | | | Stage-I | | | Ratio of Stage-II to Stage-I |
|---|---|---|---|---|---|---|---|---|
| | | Encoder / Discriminator | Decoder / Generator | Total | Encoder / Discriminator | Decoder / Generator | Total | |
| Colored-MNIST | c-VAE | 8.6M | 1.6M | 10.2M | 24.2k | 27.1k | 51k | 0.005 |
| | c-GAN | 674.0k | 2.5M | 3.2M | 24.1k | 27.1k | 51k | 0.016 |
| | c-DM | - | 5.1M | 5.1M | - | 5.1M | 5.1M | 1.000 |
| MNIST-Correlation | c-VAE | 8.6M | 1.6M | 10.2M | 150k | 150k | 300.2K | 0.030 |
| | c-GAN | 663.7k | 2.3M | 3M | 67k | 67k | 133.6K | 0.044 |
| | c-DM | | 5.1M | 5.1M | - | 5.1M | 5.1M | 1.000 |
| Corrupted-CIFAR10 | c-GAN | 668.9k | 239.5k | 3.1M | 78.6k | 78.7k | 157.2k | 0.051 |
| | c-DM | - | 9.9M | 9.9M | - | 9.9M | 9.9M | 1.000 |
| CelebA | c-DM | - | 23M | 23M | - | 23M | 23M | 1.000 |

# E  EFFECT OF TWO-STAGE TRAINING

Two-stage training of $g_{\text{mjr}}$ and $g_{\text{res}}$ instead of single-stage training is beneficial for the quality of majority generations and overall success (correctness, FID, coverage, and density) of minority generations. We present visual examples of generations in Fig. 7.

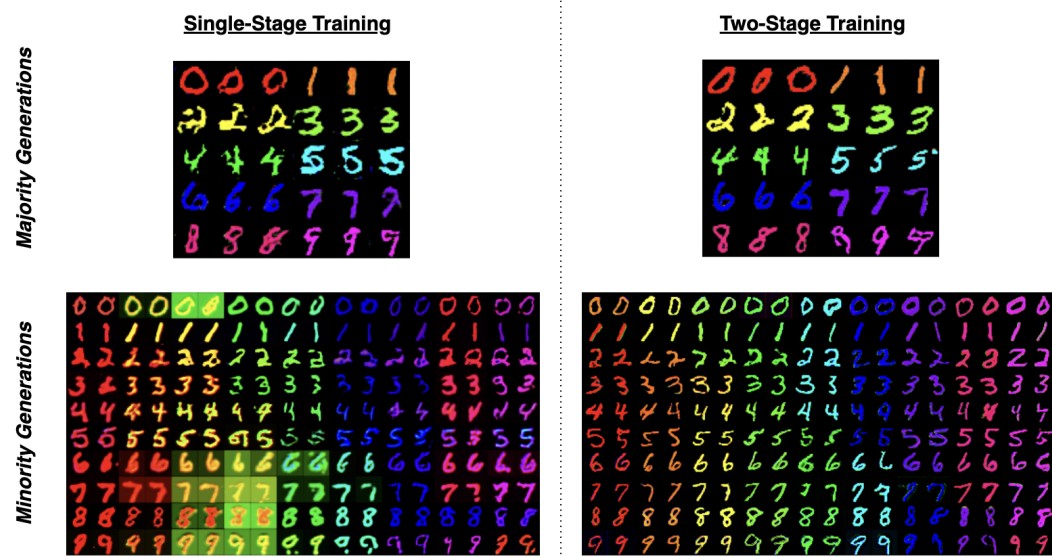

Figure 7: Sample majority and minority generations with two-stage training of $g_{\mathrm{mjr}}$ and $g_{\mathrm{res}}$ *vs.* single-stage training for 95% correlation ratio in Colored-MNIST.

## F    EFFECT OF ASSUMPTION OF CAUSAL DIRECTION

We present the comparison of generated majority and corresponding minority samples when considering causal direction $x_1 \rightarrow x_2$ *vs.* $x_2 \rightarrow x_1$ in Fig. 8. We consider ProReGen-GAN to present our analysis.

In Fig. 8-B, we simply invert the causal direction, while employing the same additive formulation for $\mathbf{y}_{\mathrm{mnr}}$ as in our main experiments and keeping the network architecture style for $g_{res}$ consistent.

We further experiment with the general formulation $\mathbf{y}_{\mathrm{mnr}} = g_{\mathrm{res}}(\mathbf{h}_{\mathrm{mjr}}(\mathbf{x}_1), \mathbf{x}_1, \gamma)$ using: 1) the same network architecture design for $g_{res}$ as in Fig. 8-B, and 2) ResNet-style architecture design for $g_{res}$ such that the residual operation occurs implicitly within the network, to assess their potential benefit for the residual generation task with the inverted causal direction *color* $\rightarrow$ *digit*. However, no noticeable improvement was observed as shown in Fig. 8C-D.

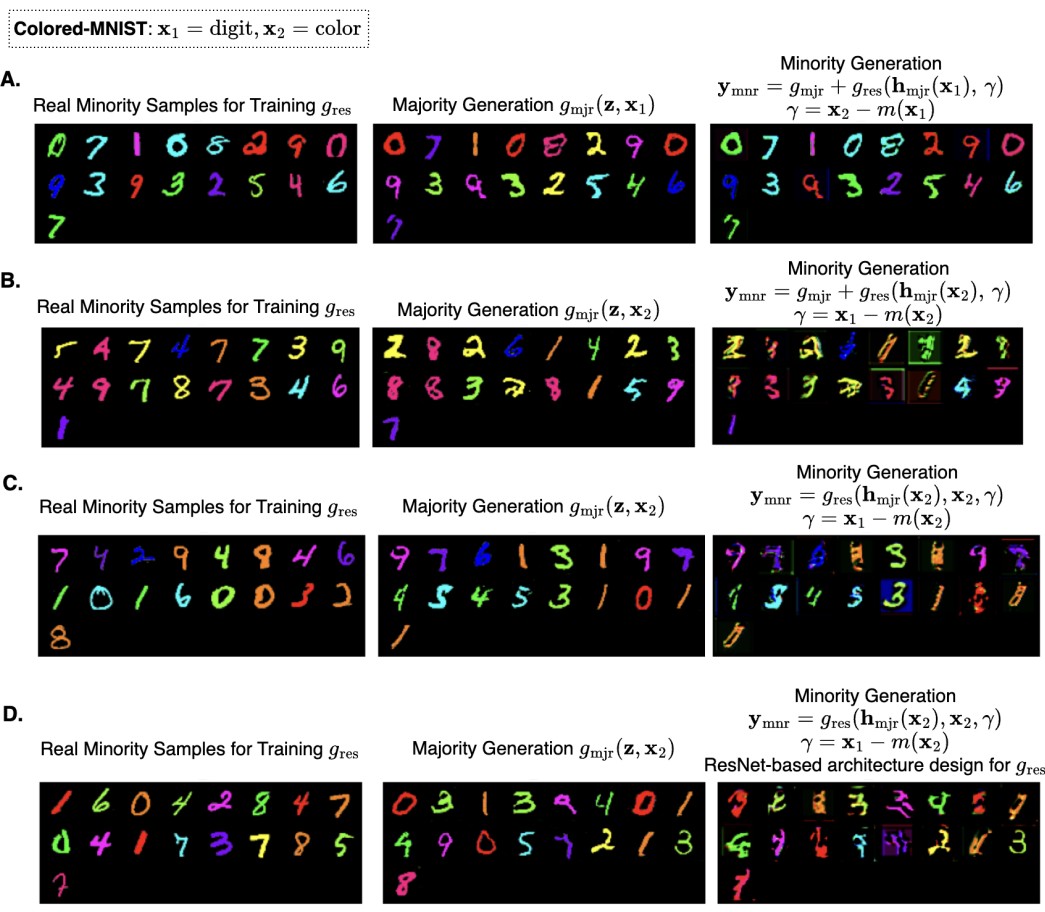

Figure 8: Examination of the effect of causal direction on the residual generation task for ProReGen-GAN, considering 95% correlation ratio in Colored-MNIST. Real minority samples used for training $g_{\text{res}}$ are shown in *left* with corresponding majority (middle) and minority (right) generation samples. We employ the causal direction *digit* $\rightarrow$ *color* for Colored-MNIST in our main experiments and present the sample results in **A**. We explore the effect of inverting the causal direction to *color* $\rightarrow$ *digit* in **B**, **C**, and **D**, where the roles of $\mathbf{x_1}$ and $\mathbf{x_2}$ are reversed.

Table 12: Sensitivity of ProReGen to errors in estimation of $m(\mathbf{x}_1)$ for GAN trained on Colored-MNIST with 95% correlation ratio

| Perturbation Level | Overall Correctness ↑ | Digit Correctness ↑ | Color Correctness ↑ | FID ↓ | Coverage ↑ | Density ↑ |
|---|---|---|---|---|---|---|
| 0% | $0.9256 \pm 0.0257$ | $0.9490 \pm 0.0053$ | $0.9718 \pm 0.0228$ | $17.2562 \pm 7.8816$ | $0.7519 \pm 0.0903$ | $0.6089 \pm 0.1216$ |
| 50% | $0.8909 \pm 0.0124$ | $0.9579 \pm 0.0009$ | $0.9291 \pm 0.0124$ | $13.6764 \pm 3.7443$ | $0.8341 \pm 0.0344$ | $0.6292 \pm 0.0468$ |
| 80% | $0.8589 \pm 0.0139$ | $0.9541 \pm 0.0007$ | $0.8974 \pm 0.0117$ | $13.4902 \pm 1.4889$ | $0.8349 \pm 0.0252$ | $0.6237 \pm 0.0089$ |

# G  Effect of Error in Estimation of $m(\mathbf{x}_1)$

We present the sensitivity of ProReGen to errors in estimation of $m(\mathbf{x}_1)$ for GAN trained on Colored-MNIST with 95% correlation ratio in Table 12. As described in the main text, we experimented with three levels of perturbation with increasing percentage of samples induced with random shifting in attribute $\mathbf{x}_1$ per training epoch. As shown, for ProReGen-GAN trained on Colored-MNIST with 95% correlation strength, the correctness of minority generation dropped as expected with the increase in the level of perturbation, although not rapidly and still improved over the naive model at 80% of errors.

# H COMPUTATION COST

Table 13 lists the training time on Colored-MNIST with 98% correlation strength by naive, re-weighting, and ProReGen models for c-VAE, -GAN, -DM. It highlights the benefit inherent in the design choice of ProReGen: the minimal overhead of computation of stage-II training in comparison to stage-I and naive models ($\sim 2\%$ for all models) .

Table 13: Seconds per epoch for c-VAE, c-GAN, and c-DM for training on Colored-MNIST at 98% correlation strength. The experiments were performed on a single NVIDIA RTX 2080 Ti GPU with 10.75 GB VRAM and batch size 32.

|  | Naive | Re-weighting | Stage I | Stage II |
|---|---|---|---|---|
| **c-VAE** | $8.9442 \pm 0.1116$ | $8.9432 \pm 0.0610$ | $8.6888 \pm 0.0102$ | $0.1863 \pm 0.0003$ |
| **c-GAN** | $16.5894 \pm 0.0142$ | $16.6545 \pm 0.0479$ | $13.4155 \pm 0.0379$ | $0.2676 \pm 0.0013$ |
| **c-DM** | $121.9143 \pm 5.6574$ | $125.2077 \pm 1.4170$ | $125.7700 \pm 0.0962$ | $2.9127 \pm 0.0116$ |

# I    CONVERGENCE RESULTS

Fig. 9 provide examples of loss curves for ProReGen-VAE, -GAN, and -DM when trained on Colored-MNIST at correlation strength of 95%, showing stable convergence that were typical of the experiments we observed.

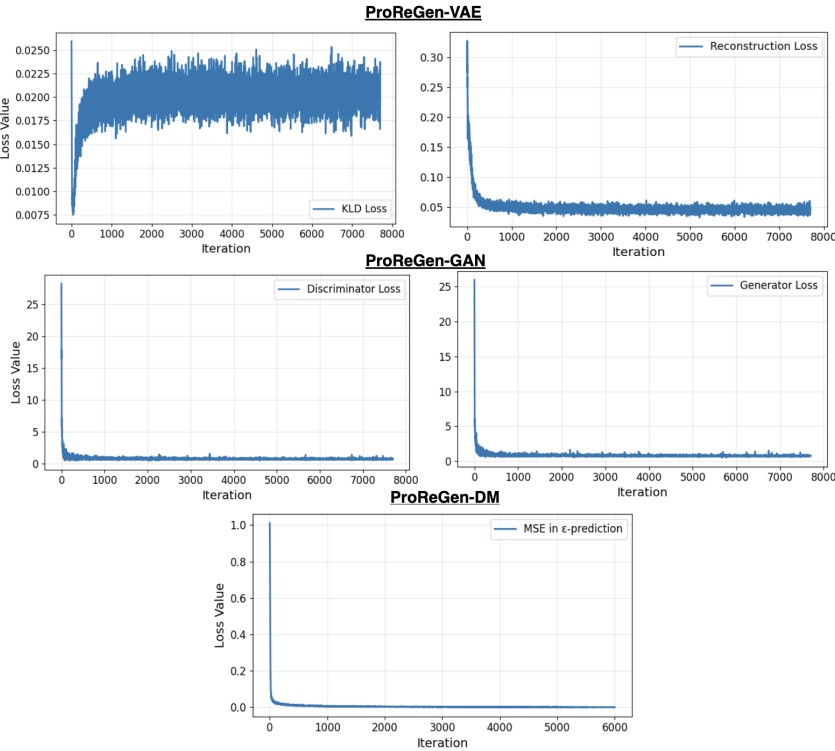

Figure 9: Examples of loss curves for ProReGen-VAE, -GAN, -DM.

## J  PERFORMANCE SENSITIVITY TO THE SIZE OF RESIDUAL SUB-NETWORK

We present a small example illustrating the sensitivity of ProReGen to the number of convolution-blocks in residual sub-net in Table 14. We can observe that both the correctness and quality of generations are worse when limiting the size of the residual sub-network to a single convolution-block. Visual examples in Fig. 10 further demonstrate the issue.

Table 14: An example from ProReGen-GAN trained on 95% correlation strength in Colored-MNIST illustrating the sensitivity of performance to the size of the residual sub-network. The metric values are for minority generations.

|  | Correctness ↑ | FID ↓ | Coverage ↑ | Density ↑ |
|---|---|---|---|---|
| **Two Convolution Blocks** | $0.9256 \pm 0.0257$ | $17.2562 \pm 7.8816$ | $0.7519 \pm 0.0903$ | $0.6089 \pm 0.1216$ |
| **Single Convolution Block** | $0.6476 \pm 0.0536$ | $20.6536 \pm 1.6247$ | $0.6787 \pm 0.0323$ | $0.4399 \pm 0.0476$ |

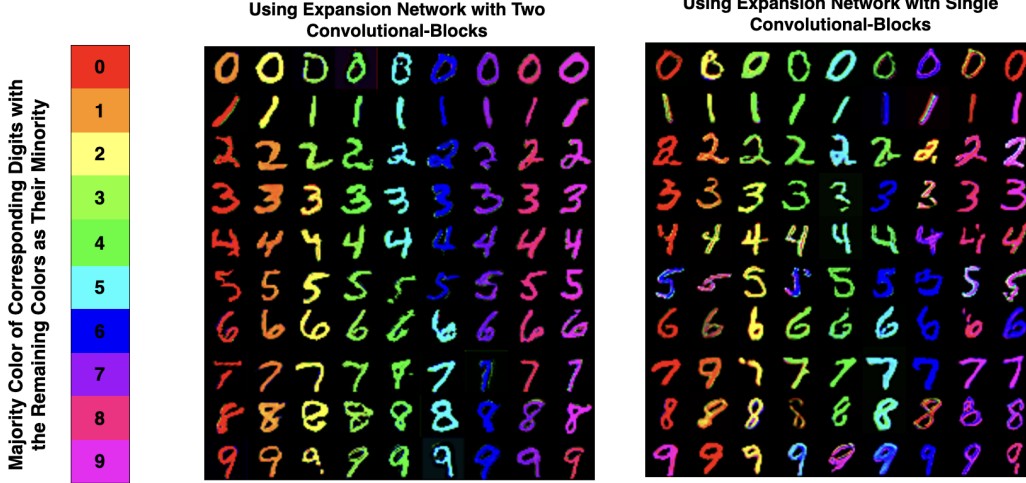

Figure 10: Visual demonstration of the sensitivity of ProReGen to the number of convolution-blocks in residual sub-net. The generations were obtained from ProReGen-GAN trained on Colored-MNIST at 95% correlation strength and were intended to yield minority colors per row. We can observe that almost all generations match the intended color when employing two convolutional-blocks while using a single convolution-block yielded greater number of error cases.

## K    ADDITIONAL RESULTS

### K.1    COLORED-MNIST

We present the comparison of FID and density metric values of ProReGen against the baselines in Fig. 11, Fig. 12, and Fig. 13.

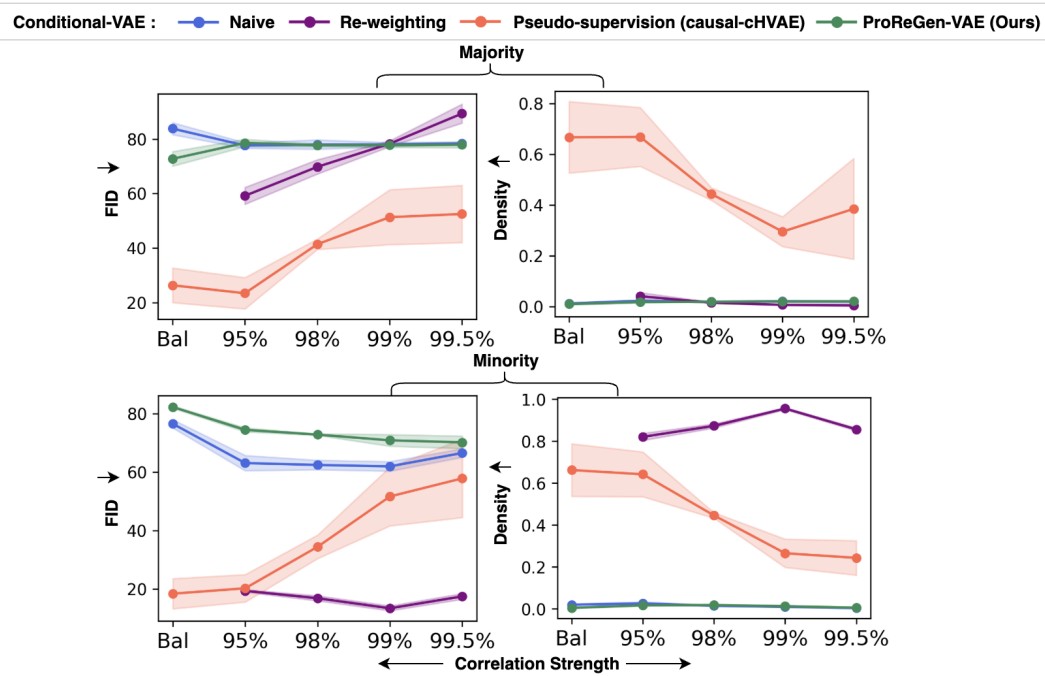

Figure 11: Comparison of FID and density metric values of ProReGen-VAE against the baselines for Colored-MNIST dataset.

### K.2    MNIST-CORRELATION

We present the comparison of coverage and density metric values of ProReGen against the baselines for MNIST-Correlation in Fig. 14 and Fig. 15.

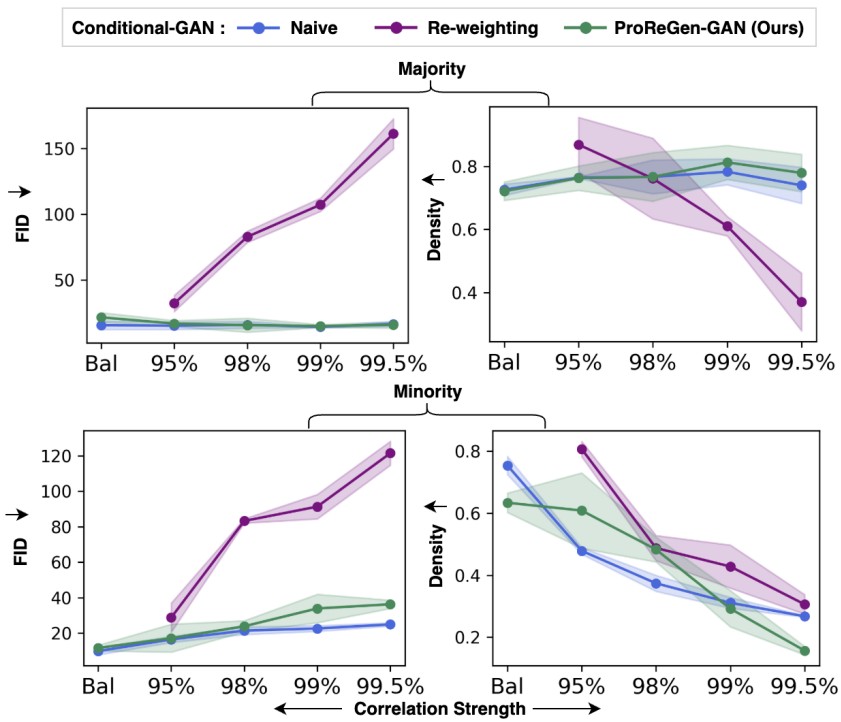

Figure 12: Comparison of FID and density metric values of ProReGen-GAN against the baselines for Colored-MNIST dataset.

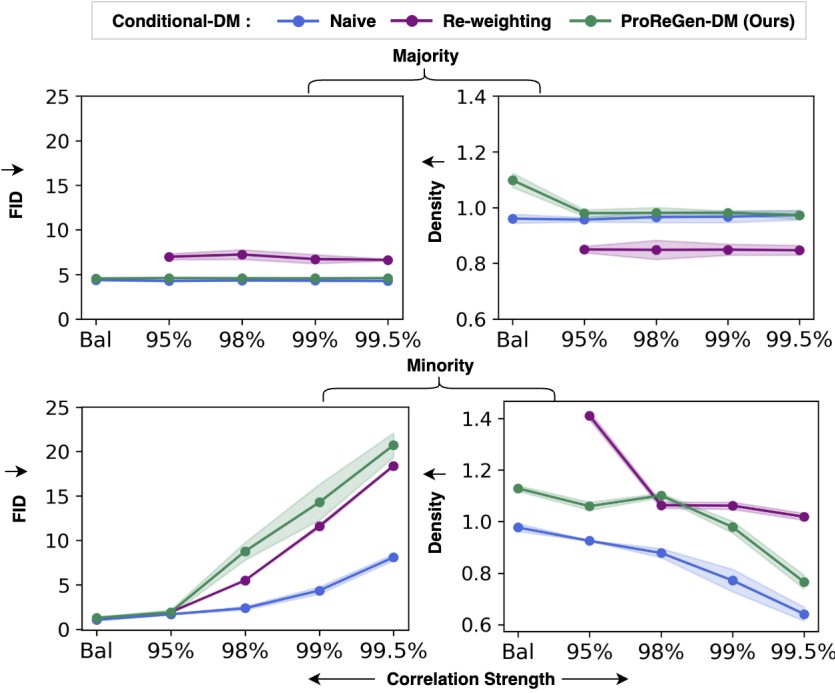

Figure 13: Comparison of FID and density metric values of ProReGen-DM against the baselines for Colored-MNIST dataset.

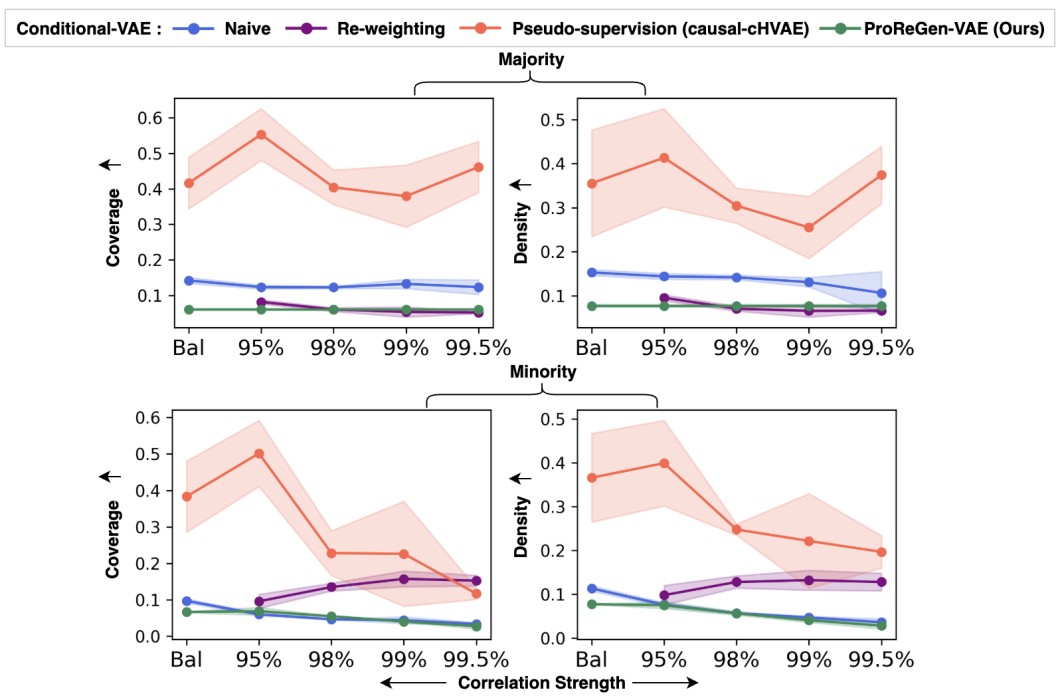

Figure 14: Comparison of coverage and density metric values of ProReGen-VAE against the baselines for MNIST-Correlation dataset.

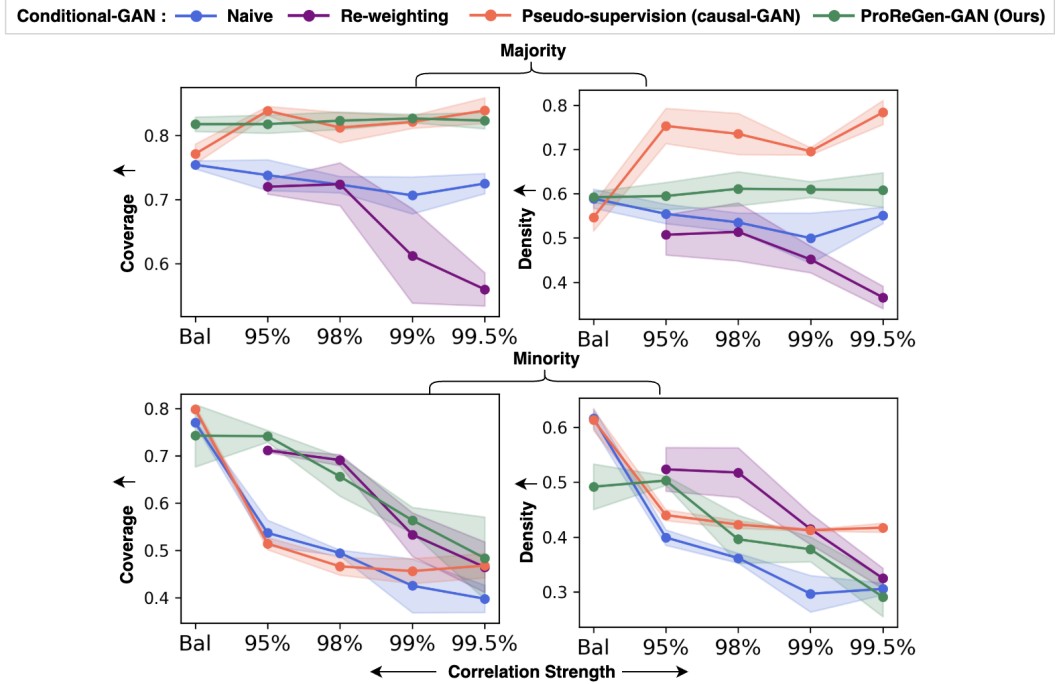

Figure 15: Comparison of coverage and density metric values of ProReGen-GAN against the baselines for MNIST-Correlation dataset.

### K.3 CORRUPTED-CIFAR10

We present the comparison of FID and density metric values of ProReGen against the baselines for Corrupted-CIFAR10 in Fig. 16 and Fig. 17. Due to limited oracle dataset (260 per subgroup), we

only have $5 \times 260 = 1,300$ total majority oracle samples to compare against. For FID calculation (using Seitzer (2020)), we require at least $2,500$ samples. Therefore, we are unable to calculate FID for majority generations, and present the FID metric values only for minority generations where we have $20 \times 260 = 5200$ total oracle samples to compare against.

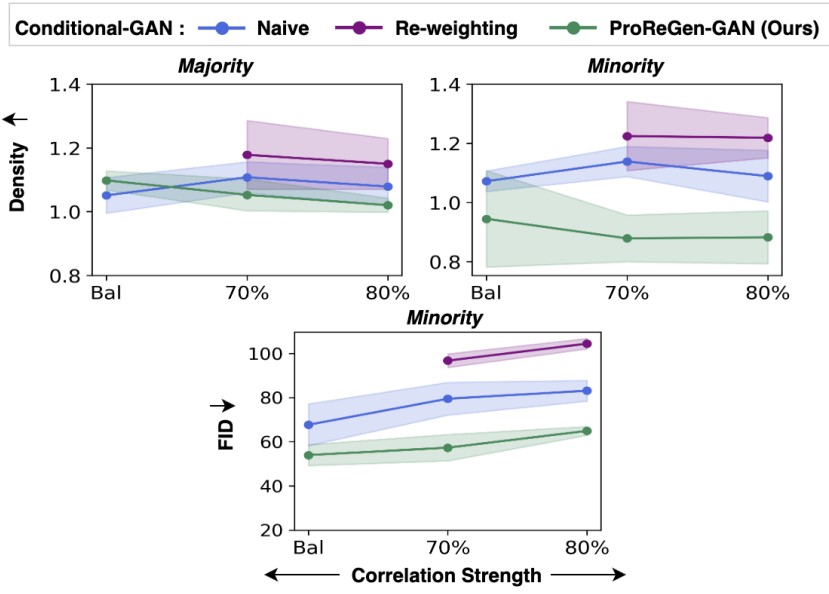

Figure 16: Comparison of FID and density metric values of ProReGen-GAN against the baselines for Corrupted-CIFAR10 dataset.

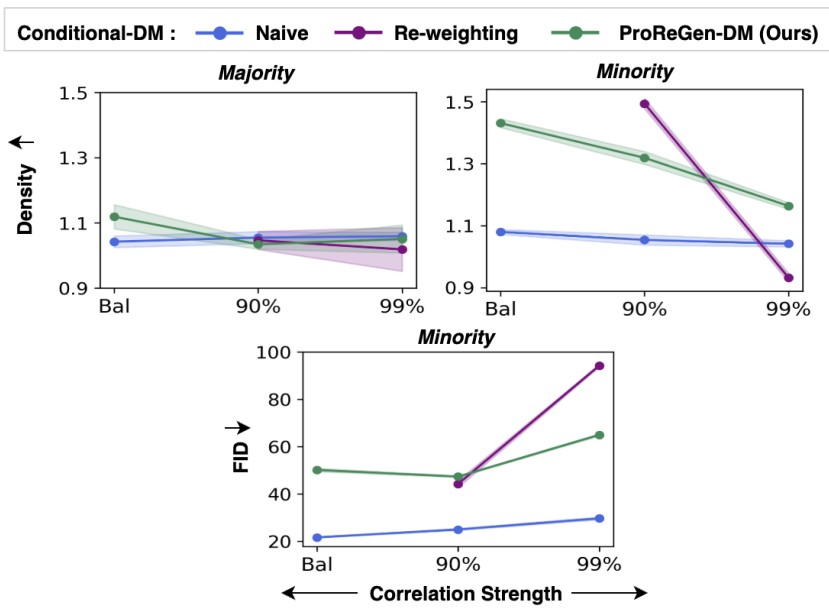

Figure 17: Comparison of FID and density metric values of ProReGen-DM against the baselines for Corrupted-CIFAR10 dataset.

### K.4   CELEBA

We present additional visual results of minority image generations on CelebA with natural correlation (Fig. 18) and reduced proportion of female with black hair color to match that of male with blond hair color (Fig. 19).

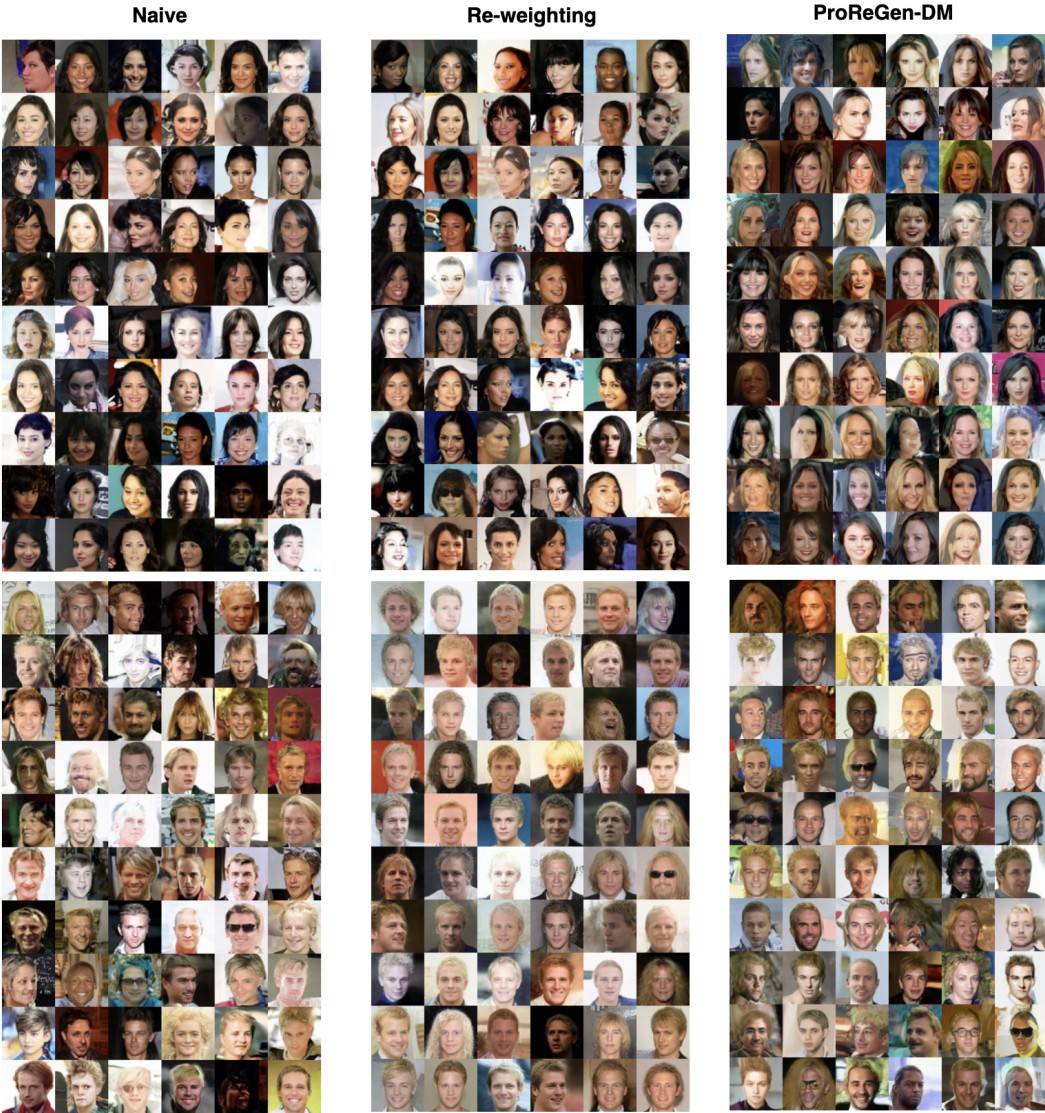

Figure 18: Comparison of naive c-DM, re-weighting, and ProReGen-DM on minority image generation on CelebA with natural correlations between gender and hair color.

**Naive**                    **Re-weighting**                    **ProReGen-DM**

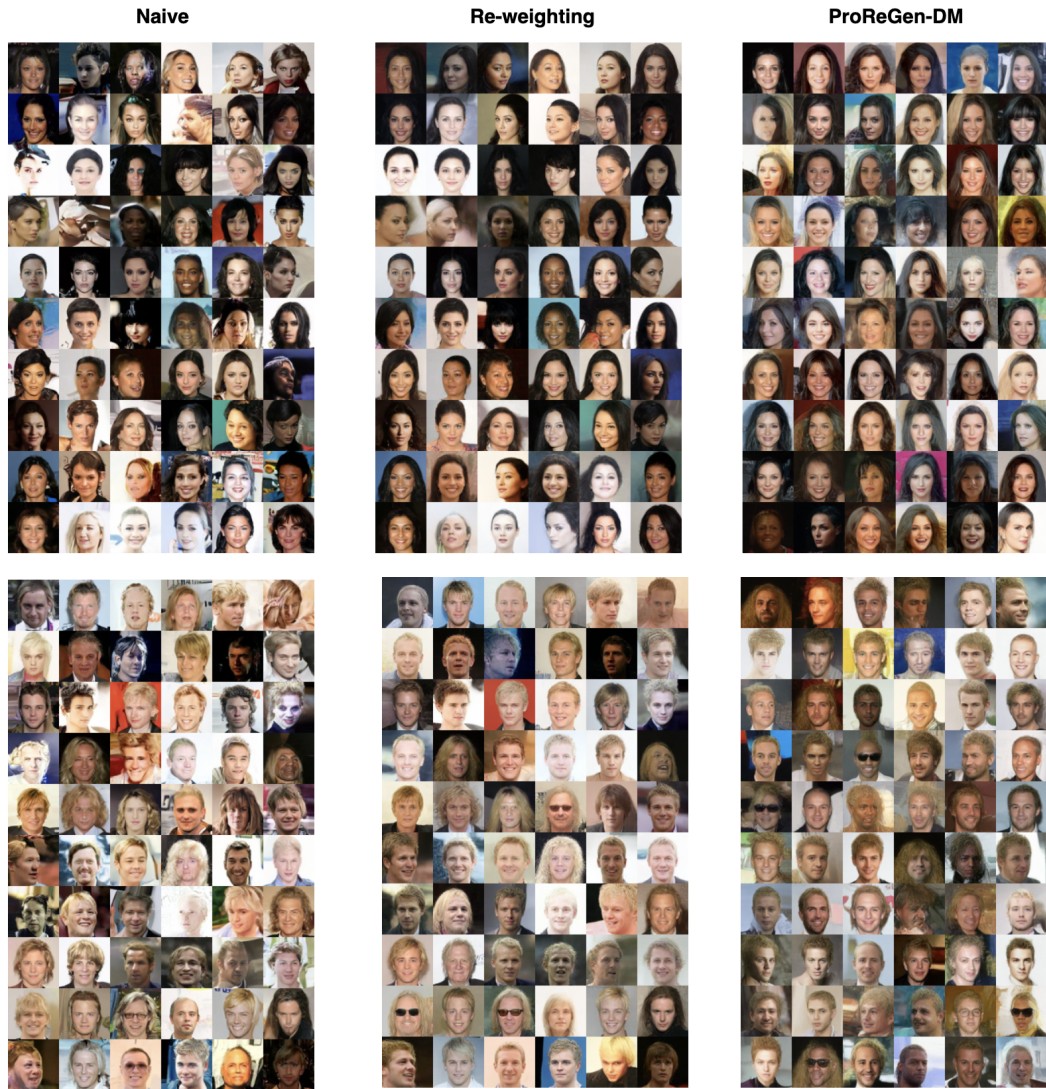

Figure 19: Comparison of naive c-DM, re-weighting, and ProReGen-DM on minority image generation on CelebA with 94% correlations between gender and hair color.

## L   LLM USAGE

We used the LLM tool, ChatGPT, at limited capacity. ChatGPT was leveraged for improving the quality of sentences to provide better readability and for grammatical corrections. Moreover, we utilized it to generate some portions of the graph creation scripts.

