# OpenReview forum: "ProReGen: Progressive Residual Generation under Attribute Correlations"
_ICLR.cc/2026/Conference — ICLR 2026 Poster_

### Official Review · Reviewer_eCdH · 2025-10-30

**Soundness:** 2
**Presentation:** 3
**Contribution:** 2
**Rating:** 6
**Confidence:** 3

**Summary:**

The authors study the problem of training a generative model under attribute correlation. To this end, they propose a novel approach that separates training into two stages. First, a part of the model is trained on the majority data, while in the second stage the network learns only the residual information for the correlated attribute. The efficacy of the method is evaluated on variations of MNIST and CIFAR.

**Strengths:**

- The main idea is novel and clearly presented.

- The studied problem is relevant and the proposed solution appropriately motivated.

- The experimental results highlight the efficacy of the method in variations of MNIST and CIFAR10.

- I appreciate the ablation of the inverse causal direction for MNIST.

**Weaknesses:**

- The main weakness of the paper lies in the insufficient experimental support. To showcase the effectiveness of the method, as well as the importance of the task, I would expect evaluation on more complex and natural scenarios. For example, a multi-attribute dataset e.g., [1], where this kind of attribute imbalances are naturally occurring would be appropriate.

- I am missing a discussion on earlier works that study generative models under imbalanced attribute distributions. For example,  [2],  [3] to name a few.

- It would be interesting to discuss how (or whether) the presented method can be applied to SOTA image generation models e.g., diffusion-based models.

- On a similar note, I am missing a discussion on the form of the studied problem in scenarios, e.g., text-to image,  where the conditioning variable lives in a combinatorially large space. How would imbalances affect the performance in such models?

- It would be valuable to discuss the presented method under the light of disentanglement and potentially add relevant comparisons.

[1]. Deep Learning Face Attributes in the Wild

[2]. Bias and generalization in deep generative models: An empirical study.

[3]. Multilinear Latent Conditioning for Generating Unseen Attribute Combinations

**Questions:**

I would appreciate if the authors address the main points raised in the weaknesses section. In particular, I would encourage further experimentation on multi-attribute images (e.g., [1]). Further discussion/comparison to earlier works on disentanglement and generative modelling under imbalanced attribute distributions would also improve the paper.

---

> ### Author Response · Authors · 2025-11-26
> **Response to Reviewer eCdH**
>
> **Addition of ProReGen on diffusion models and CelebA dataset**
>
> We are happy to report that the revised manuscript now includes additional results of ProReGen on both diffusion models, and CelebA as an example of realistic multi-attribute  vision data with naturally occurring correlations. We refer the reviewer to our overall response points 1-2 for additional details.
>
>
> **Discussion of earlier works that study generative models and disentanglement under imbalanced attribute distributions**
>
> We thank the reviewer for the suggested related works, which prompted us to add a broader literature review in the revised manuscript to clarify the distinction of ProReGen to these prior works. We refer the reviewer to our overall response point 3 for additional details.
>
>
> **Discussion of scenarios where the conditioning variable lives in a combinatorially large space**
>
> This is an excellent suggestion by the reviewer. We expect attribute correlation and imbalance to persist and even amplify in text-to-image models because the conditioning variable lives in a combinatorially large space. As a result, most semantic attributes—and particularly their combinations—appear with extremely low frequency in the training distribution. This makes imbalance significantly more severe than our current problem setting. Moreover, linguistic attributes exhibit strong co-occurrence patterns (e.g., demographic descriptors, object-action pairs), which induces correlations in the conditioning space analogous to the image-attribute correlations we study. Prior work has shown that text-to-image models struggle with rare or long-tail tokens, compositional attributes, and atypical attribute conjunctions. Because of the sparsity and heavy-tailed nature of text-based conditioning, we expect this to be an exciting avenue for future work and added such discussion in Section 6 of the revised manuscript.

---

### Official Review · Reviewer_P42c · 2025-11-03

**Soundness:** 2
**Presentation:** 3
**Contribution:** 2
**Rating:** 6
**Confidence:** 4

**Summary:**

ProReGen is an approach for generative modeling that addresses the challenge of generating underrepresented minority samples in the presence of attribute correlations. The paper proposes a two-stage progressive learning framework inspired by Robinson's partialling-out transformation, which orthogonalizes correlated input attributes and decomposes generation into majority and residual components. The method has been evaluated on Colored-MNIST, MNIST-Correlation, and Corrupted-CIFAR10, showing improvements in minority sample generation correctness compared to naive baselines and existing mitigation strategies

**Strengths:**

- The paper demonstrates both VAEs and GANs, which is diverse in nature
- The application of partialling out transformation for generative modelling is well motivated
- The paper evaluates across multiple metrics and provides multiple ablation studies
- The existing methods rely on signals from an external classifier to provide pseudo-supervision to the conditional generative models. Interestingly, they partially out an image from its components

**Weaknesses:**

- The paper demonstrates both VAEs and GANs, which is diverse in nature
- The application of partialling out transformation for generative modelling is well motivated
- The paper evaluates across multiple metrics and provides multiple ablation studies
- The existing methods rely on signals from an external classifier to provide pseudo-supervision to the conditional generative models. Interestingly, they partially out an image from its components

**Questions:**

How sensitive is the performance to eros in estmaiting m(x1)?
Why does cGAN not perform well compared to the original paper
What happens when the oracle classifiers are not so accurate
Can the framework also incorporate continuos attributes or only discere ones?

---

> ### Author Response · Authors · 2025-11-26
> **Response to Reviewer P42c**
>
> **Sensitivity to errors in m(x1)**
>
> In Section 5.5 of the revised manuscript, we added an ablation study on the performance of ProReGen versus the errors in m(x1). This was done by perturbing the label of x1 via a uniform random shift, applied to an increasing percentage of samples at the range of 0-80%. The results showed that ProReGen’s correctness in minority generation decreased as this error increased, but not rapidly and was still able to outperform naive baseline at 80% errors.
>
> **Performance of cGAN**
>
> The datasets considered in the original cGAN paper were very different from what we considered. Most importantly – in our paper, cGAN was tested under varying strengths of **attribute correlations** and the generation performance was reported on **minority images**: these were not considered in the original cGAN paper, and may have contributed to the performance difference noted by the reviewer.
>
> In the CIFAR dataset that was considered in both the original cGAN paper and ours, three key differences exist: 1) the original c-GAN paper used DCGAN as a feature extractor that was pretrained on ImageNet, whereas we did not consider such pre-training; 2) we considered corrupted versions of CIFAR; and 3) there were correlation between corruption types and object classes in corrupted-CIFAR. These differences most likely resulted in the performance difference noted by the reviewer on this particular dataset.
>
> **Accuracy of oracle classifier**
>
> The reviewer asked an excellent question that points to one of the most significant challenges in the evaluation of minority image generation. Imperfection in the oracle classifier will definitely affect the evaluation metrics, while in reality a perfect oracle is generally not possible. In image data with natural attribute correlations, for instance, it would be very difficult to establish an oracle classifier on a “balanced” version of the dataset without such correlation – this was also why we chose to use datasets with synthetically curated levels of attribute correlations in the original submission. Finally, since the oracle can only be trained on real data, its performance on generated image samples --- across a range of potentially varying generation qualities --- can also be affected even if it could be perfect on real data. In the revised manuscript, we added this discussion to Section 6 of the revised manuscript.
>
> **Continuous versus discrete attributes**
>
> The methodology of ProReGen is directly applicable to continuous attributes – it may actually better reflect the distribution of the residual if the attributes are continuous in nature. Our current methodology in training ProReGen in two stages does assume the majority/minority subgrouping of image samples given their attributes – as long as this assumption is met, ProReGen as presented is directly applicable regardless whether the attributes are discrete or continuous in nature.
>
> As the next step of ProReGen, we will extend it to setting where knowledge of these subgroups does not exist – in such setting, both stage I and stage II will be trained on all image samples, and we will leverage their attribute residual to determine the extent to which an image sample follows the attribute correlation and use it to modulate their roles in the learning in each stage. We added this discussion to Section 6 of the revised manuscript.

---

### Official Review · Reviewer_dzuq · 2025-11-03

**Soundness:** 2
**Presentation:** 3
**Contribution:** 3
**Rating:** 4
**Confidence:** 3

**Summary:**

This paper proposes a new approach for training deep generative models (DGMs) that are robust to under-represented attribute combinations. Given attributes X_1 and X_2, the model learns g(X_1, $\gamma$) rather than g(X_1, X_2), where $\gamma$ is the residual after predicting X_2 given X_1. This, the generative model is conditioned on orthogonal variables, rather than potentially very correlated variables. This makes the model more robust when generating rare attribute combinations.

**Strengths:**

- The proposed approach of orthogonalizing attributes and the two-stage residual training approach is novel. The proposed solution is intuitive and seems very reasonable.

- The fact that ProReGen is agnostic to the choice of underlying DGM, rather than being designed with only VAEs or the like, is a clear strength of the proposed approach

- The writing is clear and easy to follow

**Weaknesses:**

- The experiments are all on toy, synthetic or partially-synthetic data. The evaluation is performed on colored-mnist, mnist correlation, and Corrupted-CIFAR10 (wherein synthetic noise was added to CIFAR images). This is not a very convincing evaluation; at least, it does not provide evidence that the proposed ProReGen approach would correctly model under-represented attribute combinations in realistic data with natural correlations.

- The authors claim that the proposed approach is agnostic to the choice of deep generative models, but only discuss and implement it on VAEs and GANs. I am surprised to not see results or discussion on the case where Diffusion models (DDPM, DDIM, latent diffusion, etc) are used as the base DGM.  I don't see a technical reason why diffusion wouldn't work; the authors' claim that the choice of DGM is arbitrary seems reasonable to me.  For that reason, I think the authors should either empirically evaluate PreReGen on Diffusion models, or else clarify why this is not possible and then amended their claim about the DGM being model agnostic.

- This is a minor point, but I believe the paper would benefit if it included a discussion of property-controllable VAEs [1]. PCVAE is definitely different, as it tries to learn a disentangled latent representation rather than making the model robust to correlated attributes and minority samples. But the underlying goals are similar enough that I believe comparing and contrasting the proposed approach with existing property-controllable works would be interesting.

[1] Guo, Xiaojie, Yuanqi Du, and Liang Zhao. "Property controllable variational autoencoder via invertible mutual dependence." ICLR. 2020.

**Questions:**

Can a Diffusion model be used as the underlying DGM? Does ProReGen perform well on larger, more complicated, realistic datasets?

If these points are well addressed, then I am happy to raise my score.

---

> ### Author Response · Authors · 2025-11-26
> **Response to Reviewer dzuq**
>
> **Addition of ProReGen on diffusion models and CelebA dataset**
>
> We are happy to report that the revised manuscript now includes additional results of ProReGen on both diffusion models, and CelebA as an example of realistic multi-attribute  vision data with naturally occurring correlations. We refer the reviewer to our overall response points 1-2 for additional details.
>
> **Discussion of PCVAE**
>
> We thank the reviewer for pointing us to the connection between our work and PCVAE, which we now incorporated into the discussion of related works (Section 2) in the revised manuscript. PCVAE models the latent representation as $w$ that controls property $y$ of interest and $z$ that controls the rest of the aspects in $x$. It introduces group-wise disentanglement between $z$ and $w$, along with property-wise disentanglement between $w$’s. To enable property control, an invertible function from $w_k$ to the expectation of $y_k$ is also learned. To accommodate correlated properties as a generalization of PCVAE, $w_k$ is then extended to model disentangled groups of properties with correlations allowed within the group.
>
> As pointed out by the reviewer, a major difference between PCVAE and ProReGen is that PCVAE focuses on disentangled representations, while ProReGen focus on minority image generations. Another main difference is that, while PCVAE does have an extension to allow correlated attributes with a property group (say, the k-th group), these correlated attributes with the same group are mapped to the same latent $w_k$. In other words, PCVAE is focused with the disentanglement among uncorrelated groups of properties, but not concerned with modeling the separate effect of the correlated attributes within the same property group – the latter being the focus of ProReGen. Given this different intended use, we did not add PCVAE as another baseline but incorporated it in the added discussion of related work in the revised manuscript.

---

### Official Review · Reviewer_7gY6 · 2025-11-05

**Soundness:** 2
**Presentation:** 2
**Contribution:** 2
**Rating:** 2
**Confidence:** 2

**Summary:**

The paper proposes ProReGen that tackles conditional generation modeling for synthesizing under-represented minority images. ProReGen first learns the predictable part of one attribute from another and then treats what’s left as a residual signal that highlights the minority cases. Training is performed in two stages: Stage I trains a standard VAE/GAN on the common cases; Stage II freezes this backbone and adds a small residual-conditioned module trained only on minority examples to correct the missing variations. Therefore, this separates the attributes so the base learns general realism while the add-on learns rarity (minority attributes). Both VAE and GAN applicability is shown on Colored-MNIST, MNIST-Correlation, and Corrupted-CIFAR-10. The method improves correctness on rare attribute combinations and achieves competitive performance (FID, Coverage, Density)

**Strengths:**

- The proposed approach is explained in detail and could be easy to incorporate ie (progressive training scheme and plug and play for VAE/GAN architecture).
- Proper baselines are discussed in the related section and used to compare the performance.

**Weaknesses:**

- It would be great to show the results on some real and large scale vision dataset to help the readers evaluate the efficacy of the proposed approach. It’s unclear how the gains would translate to higher-resolution, richly annotated datasets (faces, scenes) or to modern diffusion/flow models, which are the latest common practice. A larger-scale study would strengthen the empirical case.
- Stage-II adds networks and trains only on minority samples while freezing the backbone. It would help the readers to show discussions around added compute cost, convergence stability, or sensitivity to the size of the residual sub-network.

**Questions:**

Please refer to the weakness section

---

> ### Author Response · Authors · 2025-11-26
> **Response to Reviewer 7gY6**
>
> **Addition of ProReGen on diffusion models and CelebA dataset**
>
> We are happy to report that the revised manuscript now includes additional results of ProReGen on both diffusion models, and CelebA as an example of large scale multi-attribute  vision data with naturally occurring correlations. We refer the reviewer to our overall response points 1-2 for additional details.
>
> **Computation cost, convergence stability, and sensitivity to the size of the residual sub-network**
>
> We added a table summarizing the parameter count for the stage I and stage II model of ProReGen in Appendix C of the revised manuscript (Table 11 in Appendix D), showing the overhead of the expanded residual sub-network over the naive baselines that adopt the architecture of the stage I model. In Appendix H of the revised manuscript, we added a comparison of computational cost of ProReGen’s stage-I and stage-II training versus naive/reweighting baselines, using Colored-MNIST with 98% correlation strength as an example. It provides evidence that stage II training incurred minimal overhead over stage I or naive training (2% computation of stage I).  We in general did not observe convergence instabilities in ProReGen, and added examples of loss curves representative of ProReGen-VAE, -GAN, and -DM in Appendix I of the revised manuscript.
>
> In our observations, the appropriate size of the residual sub-network mainly depends on the base stage-I architecture, the dataset, and the difficulty of the residual tasks: e.g., four conv-blocks were used in the residual sub-net in MNIST-Correlation to add or remove the zigzag, versus two conv-blocks were sufficient in Colored-MNIST for changing the color of the digit. In Appendix J of the revised manuscript, we added a small experiment to test the sensitivity of ProReGen to the number of conv-blocks in residual sub-net on ColoredMNIST, which showed that both the correctness and quality of generations were worse when limiting the size of the residual sub-network to a single convolution-block. In general, this design choice should be informed by knowledge about the dataset and generation task complexity.

---

### Author Response · Authors · 2025-11-26
**Summary of Major Revision**

We thank all reviewers for their support and constructive feedback. Below we summarize the major changes included in the revised manuscript, in response to previous comments shared across reviewers. Additional responses to individual reviewers are posted in their respective sections.

**Addition of ProReGen on diffusion models** (Reviewer 7gY6,dzuq,eCdH): We extended the concept of ProReGen to the conditional DDPM model, and evaluated it on all datasets and correlation strengths considered in our paper (except in MNIST-correlation because the highest level of correlation did not induce a bias in the naive c-DDPM model). The methodology details are added to **Section 4.4** in the revised manuscript, and experiments results to **each subsection of Section 5** for the relevant dataset. In brief, the first-stage model is a naive conditional-DDPM conditioned on x_1 and trained on majority images only. The second-stage model extends this majority c-DDPM by a copy of the same c-DDPM to denoise a minority image, but now 1) conditioned on the residual attribute for x_2, and 2) receiving feature injections in all layers from the frozen first-stage c-DDPM that denoises a majority image paired with the minority image. The generation results showed that, compared to naive c-DDPMs,  the design of ProReGen to leverage first-stage c-DDPM is critical for both improving the correctness of minority generation and overcoming the overfitting issues to the small number of minority training samples.

**Addition of CeleA results (natural correlations in realistic datasets)**  (Reviewer 7gY6,dzuq,eCdH): We adopted CelebA to test ProReGen on large-scale realistic multi-attribute vision datasets with naturally-occurring attribute correlations. Because it was not possible to train an accurate oracle classifier on this type of datasets (where a “balanced” version does not exist), the experimental results were presented qualitatively for visual inspection in **Section 5.4** of the revised manuscript. We considered only DM-based models in these experiments due to their stronger performance in earlier datasets. The results demonstrated the vulnerability of the naive c-DM to natural attribute correlations, and the feasibility of ProReGen-DM to correct such correlations in minority image generation.

**Additional discussion of related works** (Reviewer dzuq and eCdH): We thank the reviewer for the suggested related works, which prompted us to refresh and broaden our literature survey. We added the discussion of these related works, including and beyond those suggested by the reviewer, in **Section 2** of the revised manuscript under “**Disentanglement/generative modeling under attribute correlations**”. We mainly included two areas of works, including 1) works that are focused on disentangled representations, which differ from ProReGen in that they are not concerned with modeling the separate effect of the correlated attributes on the generation process, and 2) generative modeling considering rare attributes and imbalance, which are related to minority attributes but pose very different challenges to the generative models:  rare attributes present a more general generalization challenge for the model to generalize beyond sparse samples to represent infrequent/rare attributes; by contrast, attribute correlations require methods to explicitly handle dependence among co-occurring attributes as a specific form of distribution shift.

---

### Meta-Review · Area_Chair_r7jC · 2026-01-08

**Summary:**

This paper proposes ProReGen, which mitigates the issue when attribute correlation in training data harms a generative model's ability to synthesize minority samples. The approach adopts Robinson's transform to orthogonalize inputs, which separates the majority and residual components.

**Reviewer Concerns:**

The reviewers raised the following concerns

1. Not a modern generative model (7gY6, dzuq, eCdH): The reviewers criticize that the major experiments conducted with VAEs/GANs are no longer major generative models
2. Synthetic dataset (7gY6, dzuq, eCdH): The reviewers requested evaluation on real-world and large-scale image datasets.
3. Computational cost (7gY6): The reviewer raised concerns regarding the computational cost and stability of the proposed two-stage training
4. Related work (dzuq, eCdH): The reviewers pointed out missing discussions on disentanglement approaches.

**Reviewer Scores:**

The initial scores were 2(7gY6), 4(dzuq), 6(P42c), and 6(eCdH).

During the rebuttal phase, authors provide the following to address the concerns summarized above. To resolve concern 1, the authors implemented ProReGen on conditional DDPMs, and they show that progressive residual design is critical for improving minority sample generation. Regarding 2, the authors added experiments using CelebA to demonstrate the effectiveness on realistic multi-attribute data. Regarding 3, the authors provided a computational analysis and overhead in the Appendix, stating that the overhead is marginal. Regarding 4, the authors revised Sec. 2 to state that disentanglement works focus on representation rather than minority generation.

Overall, AC confirms that the authors provided a proper rebuttal, showing that the proposed idea can be applied to the real dataset. In addition, as stated by reviewer dzuq, such a real-world experiment would raise the score. Although the experiment using CelebA would not satisfy reviewer 7gY6's request for a 'large-scale dataset', AC confirms that it would not compromise the soundness of the proposed idea. AC recommends acceptance of the paper, but notes weak confidence due to the need for high-quality results, although the paper addresses the fundamental issue of the generative model.

---

### Decision · Program_Chairs · 2026-01-26

Accept (Poster)